# AN AUTOMATED DATA ENGINEERING PIPELINE FOR TIME SERIES CLASSIFICATION VIA TEXT EMBEDDINGS

## ABSTRACT

Data engineering pipelines are essential - *albeit costly* - components of predictive analytics frameworks requiring significant engineering time and domain expertise for carrying out tasks such as data ingestion, preprocessing, feature extraction, and feature engineering. In this paper, we propose *ADEPT*, an automated data engineering pipeline via text embeddings. At the core of the *ADEPT* framework is a simple yet powerful idea that the entropy of embeddings corresponding to textually dense raw format representation of time series can be intuitively viewed as equivalent *(or in many cases superior)* to that of numerically dense vector representations obtained by data engineering pipelines. Consequently, *ADEPT* uses a two-step approach that (i) leverages text embeddings to represent the diverse data sources, and (ii) constructs a variational information bottleneck criteria to mitigate entropy variance in text embeddings of time series data. We further establish theoretical guarantees showing that our construction maximizes mutual information while controlling predictive error, ensuring both compression and preservation of the predictive signal. ADEPT provides an end-to-end automated implementation of predictive models that offers superior predictive performance despite issues such as missing data, ill-formed records, improper or corrupted data formats and irregular timestamps. Through exhaustive experiments, we show that the *ADEPT* outperforms the best existing benchmarks in a diverse set of datasets from large-scale applications across healthcare, finance, science and industrial internet of things. Our results show that *ADEPT* can potentially leapfrog many conventional data pipeline steps thereby paving the way for efficient and scalable automation pathways for diverse data science applications.

## 1 INTRODUCTION

Data engineering pipelines are fundamental components for enabling predictive analytics on time series data in several areas such as energy Rahimilarki et al. (2022), healthcare An et al. (2023) and finance Dingli & Fournier (2017). These pipelines broadly comprise of seven sequential steps pertaining to data ingestion; data preprocessing; feature extraction; feature engineering; model training and testing followed by model deployment Raj et al. (2020). Preprocessing typically involves data cleaning mechanisms that aim to eliminate ill-formed records Felix & Lee (2019), resolve irregularities in sampling as well as impute missing values. Feature engineering and extraction steps deal with identifying features of the input time series encoding the information most relevant to the predictive analytics task at hand Lin & Tsai (2020). For best efficiency gains, it becomes necessary to carefully customize methodological frameworks used for data cleaning, feature engineering and extraction tasks with respect to the application specific domain area and data challenges. As a result, despite several advances in the field of autoML Salehin et al. (2024), automating the data cleaning, feature engineering and extraction steps remain one of the most challenging and expensive tasks across conventional data engineering pipelines due to the need for significant manual intervention and domain expertise Salehin et al. (2024). In this paper, we present ADEPT, a framework that attempts to drastically simplify data engineering pipeline complexity by applying LLM-based text embedding models on raw text representations of input time series as a precursor to the model training step.

Fundamentally, the pattern recognition capability of any predictive model is a direct consequence of capturing temporal and spatial correlations in the time series input. From an information theoretic perspective, we argue that the entropy of embeddings corresponding to textually dense raw format representation (RFR) of time series (such as CSV, HDF5 etc.) can be intuitively viewed as equivalent to that of numerically dense vector representations obtained by data engineering pipelines. As a result, LLM-based text embeddings of time series RFRs can *also potentially be seen as alternative representations of spatial and temporal correlations essential for training a predictive model*. Consequently,

ADEPT enables a significantly simpler data representation that can be used for model training while retaining its spatiotemporal aspects. Also, the *ADEPT* framework exploits text embeddings of time series RFRs to effectively leapfrog data cleaning, feature engineering and extraction steps of data engineering pipelines. In doing so, ADEPT demonstrates significant resiliency with respect to missing data, ill-formed records, improper or corrupted data formats as well as irregular timestamps.

The methodological contribution of ADEPT relies on exploiting text embedding models primarily geared for LLM use cases as the foundational building block to power time series oriented predictive analytics tasks. As a result, the ADEPT framework leverages text embedding models as a *black box*, eliminating need

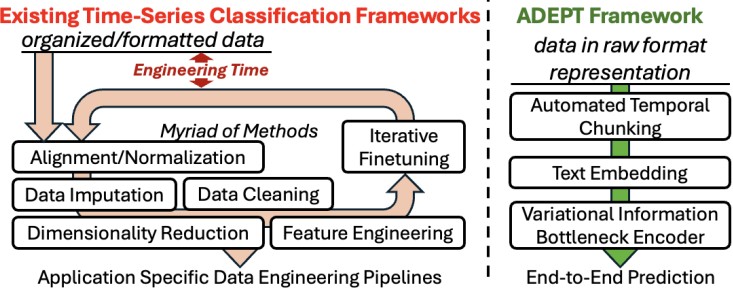

Figure 1: Comparison of the model and benchmark.

for complex fine-tuning and retraining tasks. We introduce a Variational Information Bottleneck (VIB) criteria as a means to reduce the entropy variance emanating from text embeddings of time series RFR. Additionally, the VIB criteria is used to train a multi-head attention (MHA) framework for yielding a high-quality predictive analytics model. The VIB criteria enables ADEPT to apply a filtering mechanism that relies on generating information-rich text embedding representations that can be used to boost classification accuracy. Using the VIB criteria, ADEPT can be directly applied on RFRs of time series inputs without any prior preprocessing. We note that the ADEPT framework is generally extensible and can be used in conjunction with other learning paradigms such as MLPs or SVMs. Figure 1 presents an overview of the capabilities of existing time-series classification frameworks versus the proposed *ADEPT* framework. Our contributions are as follows:

- We propose *ADEPT*, the first framework to leverage text embeddings for learning representations directly from raw time series data, enabling an automated data engineering pipeline process that is robust to data integrity issues, without finetuning or preprocessing.
- ADEPT further enhances these representations by leveraging the VIB criteria, which filters out noise and maximizes application-specific information extracted from the embeddings.
- ADEPT framework integrates of text embedding models along with VIB criteria and transformer based attention models to achieve a fully end-to-end time-series classification pipeline that rivals the performance of state-of-the-art models across diverse applications.

## 2 BACKGROUND

In this section, we formalize multivariate, multi-view classification settings that motivate ADEPT framework design, provide a review of traditional pipeline-based approaches, AutoML strategies, and recent efforts to apply text embeddings in non-text domains before introducing the information-theoretic principles for variational information bottleneck to distill noise and redundancy.

### 2.1 MULTIVARIATE TIME SERIES CLASSIFICATION

We consider a multivariate time series classification problem on a dataset of $N$ samples or events, where each event $i \in \{1, \ldots, N\}$ has an associated class label $y_i \in \{1, \ldots, C\}$ and can be observed under up to $K$ temporal views. Each temporal view is representative of a specific historical window of time series data preceding the occurrence of event $i$. Additionally, each *view* $k$ denotes a fixed window relative to some anchor point (e.g., $k = 1$ for the earliest window, $k = K$ for the most recent). Let $S_{i,k} \in \mathbb{R}^{T \times F}$ be the multivariate time series for view $k$ corresponding to event $i$ such that $S_{i,k}$ consists of $T$ timesteps and $F$ channels. In other words, we can represent the time series data for $T \times K$ consecutive time steps immediately preceding the occurrence of sample $i$ using $S_i$, where $S_i \in \mathbb{R}^{(T \times K) \times F}$ and $S_i = \texttt{Concatenate}(S_{i,k})_{k=1}^{K}$. While this formulation unifies diverse domains, practical pipelines transform $S_i$ into features via manual, domain-specific steps.

### 2.2 CONVENTIONAL TIME SERIES CLASSIFICATION PIPELINE

Conventional pipelines transform each raw series $S_{i,k}$ into features via a chain of operations that are finetuned in an iterative cycle. This process involves a series of operations, such as:

$$\tilde{S}_{i,k} = f_{\text{imp}}(S_{i,k}; \theta_{\text{imp}}), \quad \hat{S}_{i,k} = f_{\text{norm}}(\tilde{S}_{i,k}; \theta_{\text{norm}}), \quad x_i = \phi(\{\hat{S}_{i,k}\}_{k=1}^{K}; \theta_{\text{feat}}) \in \mathbb{R}^M \quad (1)$$

Here $f_{\text{imp}}$ denotes missing-value interpolation (e.g. cubic-spline McKinley & Levine (1998) or Gaussian-process Roberts et al. (2013)), $f_{\text{norm}}$ represents normalization (min–max or $z$-score), and $\phi$ extracts $M$ handcrafted features (often $M \gg TFK$; e.g. TSFEL yields $M > 9000$ Barandas et al. (2020)). Each operation depends on tuning parameters—outlier-detection thresholds $\theta_{\text{out}}$ Hodge & Austin (2004); Leys et al. (2013), dimensionality reduction technique like PCA component counts Keogh et al. (2001), or feature-selection heuristics Li et al. (2017)—leading to iterative cycles of hypothesis and validation that can occupy analysts for weeks Tawakuli et al. (2024). Inherently, a successful pre-processing and feature generation are inherently domain-specific and demand substantial engineering time and effort to tailor methods and validate results Keogh et al. (2001); Leys et al. (2013); Tawakuli et al. (2024); Li et al. (2017); Leys et al. (2013). Once these preprocessing and data-engineering steps produce the fixed-length feature vectors $\{x_i\}$, practitioners then train a classifier $g(x_i; \theta_{\text{clf}})$ to predict labels $y_i$, further extending the design burden with choices of model family and hyperparameters. *Conventional methods require significant engineering time, produce highly complex frameworks, do not generalize well across different applications and problem settings, and often require extensive retraining and parameter re-tuning when deployed in new environments.*

## 2.3 AutoML-Based Pipeline Search for Time Series Classification

To alleviate the intensive engineering effort of manual preprocessing and feature engineering, AutoML frameworks seek to automate parts of the process by formulating the problem using a joint optimization formulation over a pipeline search space $\mathcal{H}$:

$$H^* = \arg\min_{H \in \mathcal{H}} \mathcal{L}_{\text{val}}\big(\mathcal{M}(\{S_{i,k}\}; H)\big) \tag{2}$$

where $\mathcal{M}$ encompasses imputation, normalization, feature extraction, model architecture, and hyperparameters. Tools such as auto-sklearn's Bayesian optimization Feurer et al. (2015), TPOT's genetic programming Olson et al. (2016), AutoKeras's neural architecture search Jin et al. (2023), H2O AutoML's stacked ensembles LeDell & Poirier (2020), and AutoLDT's CMA-ES–driven transformer search Wang et al. (2024) have demonstrated the feasibility of AutoML methods. However, AutoML approaches also incur substantial computational overhead due to the combinatorial size of $\mathcal{H}$, often demanding days of GPU/CPU time; they produce opaque "black-box" pipelines that hinder model interpretability; they typically employ only generic imputation (e.g. mean/median) and scaling routines rather than domain-specific methods such as Gaussian-process interpolation or seasonality-aware normalization; they still rely on extensive manual filtering of large feature sets (e.g. pruning TSFEL's thousands of extracted features Barandas et al. (2020)); and their domain-agnostic search strategies frequently overlook temporal inductive biases and multi-view patterns, which can lead to suboptimal accuracy on complex sequence data. *In summary, while AutoML methods improve automation, they do not necessarily lead to a good representation of data, still require significant engineering time, and demands access to immense computational resources.*

## 2.4 Text Embedding Models for Domain-Specific Data analysis

Text embedding models—originally grounded in the distributional hypothesis Harris (1954) —treat any co-occurring entities as "tokens," unlocking cross-domain applications across spatial semantics Hu et al. (2020); Niu & Silva (2021), movement dynamics Murray et al. (2023), behavioral inference Richie et al. (2019), political discourse analysis Rheault & Cochrane (2020), joint video–text embeddings for instructional content Sun et al. (2019); Miech et al. (2019) and audio–text alignments via contrastive pretraining Guzhov et al. (2022); Ilharco et al. (2019). These foundational studies trained embeddings from scratch on domain-specific data, demonstrating that lightweight embedding architectures can effectively capture complex, domain-specific structures. Building on this legacy, modern practitioners can either deploy fully offline, open-source embedders—such as Nomic's nomic-embed-text-v1 Nussbaum et al. (2024)—for strict data residency and privacy control, or leverage API-based services like OpenAI's text-embedding-3-small OpenAI (2023), which often offer superior accuracy due to web-scale pretraining but require sending inputs to third-party servers. Because these large models are pretrained on vast, internet-scale corpora, they provide high-quality semantic vectors with far less overhead than full LLMs, streamlining experimentation without building custom models from scratch. *Text embedding models demonstrated a significant ability to capture structure across diverse domains —suggesting that even highly structured, non-linguistic data like time series may benefit from such pretrained semantic representations.*

## 2.5 Information Bottleneck Approaches in Deep Learning

Variational Information Bottleneck (VIB) techniques emerged as powerful tools in deep learning to denoise input data and enhance model accuracy. The original formulation by Alemi et al. (2016)

introduced a stochastic encoder–decoder framework that improved model robustness on MNIST and ImageNet by compressing task-irrelevant features in the latent space. This was extended by Achille & Soatto (2018), who proposed Information Dropout—a parameterized log-normal noise model that promotes invariant, disentangled representations. Foundational insights from noisy channel theory Dobrushin & Tsybakov (1962) support VIB's core mechanism of stochastic compression. In generative modeling, the $\beta$-VAE of Higgins et al. (2017) similarly enforces factorized, expressive latent codes via a constrained variational objective. A comprehensive survey by Goldfeld & Polyanskiy (2020) synthesizes these developments, framing VIB as a unifying agent. *By suppressing redundancy and irrelevant noise, VIB generates more informative and compact latent representations, and ultimately improves the performance of the downstream prediction tasks.*

## 3 METHODOLOGY

The methodological core of ADEPT relies purely on a black box text embedding model applied on decompositions of the RFRs of time series input datasets which is followed by a VIB criterion for enhancing information gain. ADEPT methodology can be broken down into four distinct steps that can be implemented in a scalable fashion, and integrated to develop two versions of the framework.

- *RFR Processing and Decomposition*: RFRs corresponding to multichannel time series input sequences are decomposed into segments of fixed content sizes and serialized for standardization.
- *Temporal Text Embeddings*: Serialized RFR temporal decompositions are processed using black-box, LLM-based language embedding models to obtain text embeddings.
- *Variational Information Bottleneck*: A variational encoder learns the latent space distribution of RFR embeddings, resulting in fused sequences to reduce noise & enhance information gain.
- *Classifier*: A transformer-based classifier captures intra- and inter-view dependencies from the fused multi-view sequences and performs final prediction.

### 3.1 RFR PROCESSING AND DECOMPOSITION

We begin by considering the tuple $(R_i, y_i)$ for each reported event $i \in \{1, \ldots, N\}$, where $R_i = \text{RFR}(S_i)$ where $S_i \in \mathbb{R}^{(T \cdot K) \times F}$ denotes the actual time series data for $K$ temporal views immediately preceding event $i$. Next, we consider the decomposition of each temporal view $S_{i,k}$ into $M$ equal-length segments or chunks of length $L = T/M$, with corresponding RFR $R_{i,k}^{(j)} = \text{RFR}(S_{i,k}^j)$. It is important to note that the decomposition scheme preserves the temporal order of data pertaining to individual time steps across as well as within multiple views. Therefore, extracting the RFR for each segment can be trivially accomplished using a simple count based query or by enforcing a content size limit (for e.g., in KBs, MBs) on each chunk.

Our approach also balances the extremes of processing the full $T \times F$ series at once—which can dilute important local patterns and incur high computational cost—and treating each timestep independently—which ignores temporal and cross-channel structure. While temporal chunking enhances downstream representations by balancing local and global dependencies, selecting the optimal number of chunks $M$ introduces a trade-off. A smaller $M$ (longer chunks) may overload downstream encoders or mix heterogeneous patterns, whereas a larger $M$ (shorter chunks) risks fragmenting temporal dependencies and increasing sequence length. Domain insight or systematic validation studies can help inform the choice of $M$ for balancing expressivity and computational tractability. However, validation studies for determining $M$ can be easily automated and implemented in a scalable fashion on account of the pipeline simplifications afforded by the text embedding models.

### 3.2 TEMPORAL TEXT EMBEDDINGS

To leverage the powerful, pre-trained semantic priors, we treat each raw time-series chunk as text, enabling off-the-shelf embedding models to capture both numeric and categorical patterns without manual intervention. We serialize each RFR chunk $R_{i,k}^{(j)} \in \mathbb{R}^{L \times F}$ to obtain $R_{i,k}^{(j),ser} \in \Sigma^*$ by concatenating timestamps and channel readings into a token sequence where $\Sigma^*$ is the model's character set and $E$ its output dimension. We then apply a frozen text-embedding function $g$:

$$\mathbf{e}_{i,k}^{(j)} = g\big(R_{i,k}^{(j),ser}\big) \ \in \ \mathbb{R}^E \tag{3}$$

**Note:** Since $g$ natively handles both numeric and textual tokens, categorical channels (e.g., flags or event types) can be embedded alongside continuous measurements in one unified string.

By applying the pretrained text embedding model on serialized time series chunks, we can derive the lower bound on the mutual information between original time series sequences and their corresponding

embeddings. Proposition 1 captures the theoretical lower bound of mutual information $I(S_{i,k}^j, \mathbf{e}_{i,k}^{(j)})$ with respect to original time-series chunk $S_{i,k}^j$ and serialized time series embeddings $\mathbf{e}_{i,k}^{(j)}$.

**Proposition 1.** *The lower bound on mutual information between the original time-series segment $S_{i,k}^j$ and its corresponding embedding $\mathbf{e}_{i,k}^{(j)}$ has the following valid lower bound:*

$$I(S_{i,k}^j; \mathbf{e}_{i,k}^{(j)}) \geq \sum_{l=1}^n \left[ \mathbb{H}(S_{l,k}^j) - H_b(p_e^{(j,l,k)}) - p_e^{(j,l,k)} \log(|\mathcal{V}| - 1) \right] \quad (4)$$

*where $i \in \{1, N\}, j \in \{1, M\}, k \in \{1, K\}$, $\mathcal{V}$ denotes the set of classes for classification, $p_e^{(j,i,k)}$ represents the token-wise prediction error and $\mathbb{H}$ is the binary entropy function.*

*Proof.* Proof provided in Appendix B.1 □

Proposition 1 provides a theoretical justification of the lower bound that is influenced by the decoder performance. If the decoder $f$ achieves low token-wise prediction error (i.e., $p_e^{(j,i,k)} \to 0$ for all $i, j, k$), then both $H_b(p_e^{(j,i,k)})$ and $p_e^{(j,i,k)} \log(|\mathcal{V}| - 1)$ vanish. In this case, the mutual information between $S$ and $E$ approaches the total entropy of the original sequence: $I(S; E) \to \sum_{i=1}^n \mathbb{H}(s_i) = \mathbb{H}(S)$.

Proposition 1 helps assess the mutual information lower bound with respect to the serialized embeddings of raw time series textual information. Proposition 1 shows that given a robust decoder, embeddings from serialized raw formats—when passed through sufficiently expressive embedding models—can capture the full entropy of the source signal contained in pure text formats. Moreover, if a text embedding model is explicitly designed and trained on large-scale time-series datasets, it can better align the embedding space with temporal patterns and dynamics.

However, pre-trained text embeddings—trained on general LLM corpora—can also introduce noise when representing precise numerical sequences and often yield very high-dimensional, redundant vectors. To address this, we design a Variational Information Bottleneck (VIB) criteria as our next step in distilling more informative, lower-dimensional representations.

### 3.3 VARIATIONAL INFORMATION BOTTLENECK CRITERIA

To reduce noise variance, redundancy and maximize the extracted information gain from high-dimensional text embeddings $\mathbf{e}_{i,k}^{(j)}$, we adopt a VIB criterion across each view, producing compact low-dimensional encoding that retain task-relevant information. To do so, we compress $\mathbf{e}_{i,k}^{(j)}$ into a $d$-dimensional code $\mathbf{z}_{i,k}^{(j)}$ by leveraging a VIB encoder Alemi et al. (2016). For each view $k$, let $\phi_k = \{W_\mu^{(k)}, b_\mu^{(k)}, W_{\log\varphi}^{(k)}, b_{\log\varphi}^{(k)}\}$ denote the VIB encoder parameters, and $\theta_k = \{W_y^{(k)}, b_y^{(k)}\}$ the linear classifier parameters. Here $d$ is the *bottleneck dimension* and $\beta > 0$ the VIB trade-off weight.

**Stochastic encoder for view** $k$: For each chunk chunk $\mathbf{e}_{i,k}^{(j)}$, we derive a low dimensional representation defined by $\mu_k, \sigma_k$ as defined in equation 5.

$$\mu_k = W_\mu^{(k)} \mathbf{e}_{i,k}^{(j)} + b_\mu^{(k)}, \quad \log \sigma_k^2 = W_{\log\varphi}^{(k)} \mathbf{e}_{i,k}^{(j)} + b_{\log\varphi}^{(k)} \quad (5)$$

In equation 5, we clip $\sigma_k$ to lie between $[-10, 10]$, and set $\sigma_k = \exp\left(\frac{1}{2} \log \sigma_k^2\right)$. Thus, the encoder can be used to represent the conditional latent space distribution based on the observed embeddings:

$$q_{\phi_k}(z \mid \mathbf{e}_{i,k}^{(j)}) = \mathcal{N}(z; \mu_k, \text{diag}(\sigma_k^2)) \quad (6)$$

**Reparameterization trick**: In order to learn the latent space distribution conditioned on the embeddings, we apply a reparameterization trick characterized by equation 7 where $\varepsilon \sim \mathcal{N}(0, I_d)$.

$$\mathbf{z}_{i,k}^{(j)} = \mu_k + \sigma_k \odot \varepsilon \quad (7)$$

The reparameterization trick ensures that the distribution of latent space can be parametrized by $\phi_k$ which can be learned using gradient descent.

**Classification head**: We augment the VIB stochastic encoder with a classification head that is parametrized by $\theta_k$ in order to map $\mathbf{z}_{i,k}^{(j)}$ to logits represented in equation 8.

$$\ell_{i,k}^{(j)} = W_y^{(k)} \mathbf{z}_{i,k}^{(j)} + b_y^{(k)}, \quad p_{\theta_k}(y_i \mid \mathbf{z}_{i,k}^{(j)}) = \text{Softmax}(\ell_{i,k}^{(j)}) \quad (8)$$

**Per-view loss**: We use a loss function to jointly train the stochastic encoder and decoder framework:

$$\mathcal{L}_k = \frac{1}{NM} \sum_{i=1}^{N} \sum_{j=1}^{M} \Big[ -\log p_{\theta_k}(y_i \mid \mathbf{z}_{i,k}^{(j)}) \Big] + \beta \frac{1}{NM} \sum_{i,j} D_{\mathrm{KL}} \big[ q_{\phi_k}(z \mid \mathbf{e}_{i,k}^{(j)}) \parallel \mathcal{N}(0, I_d) \big] \quad (9)$$

where $D_{\mathrm{KL}}\big(\mathcal{N}(\mu, \sigma^2) \| \mathcal{N}(0, I)\big) = \frac{1}{2} \sum_{\ell=1}^{d} \big( \mu_\ell^2 + \sigma_\ell^2 - \log \sigma_\ell^2 - 1 \big)$ represents the Kullback-Leibler divergence loss. The cross-entropy term ensures each view's encoder retains predictive information; the KL term, enforces compact, robust codes. We optimize each $\mathcal{L}_k$ independently via Adam (learning rate $\eta$, batch size $B$) for $E$ epochs, yielding specialized encoder–classifier parameters $(\phi_k, \theta_k)$ for $k$.

### 3.4 Transformer-based Classifier Design

For the final prediction task, we train a classifier on the latent space variates $\mathbf{z}_{i,k}^{(j)}$. While any classifier (e.g. SVM, random forest, XGBoost) could be used, we adopt a powerful multi-head attention (MHA) based Transformer to capture both intra- and inter-view dependencies in a unified model. In the step, we train a single Transformer-based model on the fused embeddings from all $k$ views to perform final classification. For each sample $i$, view $k$, and chunk $j \in \{1, \ldots, M\}$, let $\mathbf{z}_{i,k}^{(j)} \in \mathbb{R}^E$ be the embedding. We assemble these into a sequence $Z_{i,k} = \big[ \mathbf{z}_{i,k}^{(1)}, \mathbf{z}_{i,k}^{(2)}, \ldots, \mathbf{z}_{i,k}^{(M)} \big]^{\mathsf{T}} \in \mathbb{R}^{M \times E}$. Our goal is to predict the class label $y_i \in \{1, \ldots, C\}$ for each sample $i$, leveraging all k views. Therefore, we define a Transformer Autoencoder $T_\psi$ architecture with parameters $\psi$ on the following components.

**Input projection, transformer encoder & decoder and classification head:** Each $E$-dimensional row of $Z_{i,k}$ is linearly projected to $h$ via $H_{i,k}^{(0)} = Z_{i,k} W_{\mathrm{in}} + b_{\mathrm{in}}$. Stacked Transformer encoder–decoder layers map to $H_{i,k}^{(L)} \in \mathbb{R}^{M \times h}$ and reconstruct $\hat{Z}_{i,k}$. A pooled representation $\bar{h}_{i,k}$ is then mapped to logits $\ell_{i,k} = \bar{h}_{i,k} W_y + b_y$, yielding class probabilities $p_{i,k} = \mathrm{Softmax}(\ell_{i,k})$.

**Training procedure**: We first *pretrain* $T_\psi$ as a joint autoencoder by minimizing the mean-squared reconstruction error averaged across the k views as represented in equation 10.

$$\mathcal{L}_{\mathrm{AE}} = \frac{1}{K.N} \sum_{i=1}^{N} \sum_{k=1}^{K} \frac{1}{2M.E} \big\| \hat{Z}_{i,k} - Z_{i,k} \big\|_Z^2 \quad (10)$$

This encourages the model to learn a latent representation that reconstructs all fused embeddings, capturing common structure across views. Next, we *fine-tune* for classification by computing the per-view distributions $p_{i,1}, p_{i,2}, p_{i,3}$ for each sample $i$, leading to a consensus given by equation 11.

$$p_i = \frac{p_{i,1} \odot p_{i,2} \odot p_{i,3}}{\sum_{c=1}^{C} \big[ p_{i,1} \odot p_{i,2} \odot p_{i,3} \big]_c} \quad (11)$$

and minimize the negative log-likelihood loss $\mathcal{L}_{\mathrm{NLL}} = -\frac{1}{N} \sum_{i=1}^{N} \log p_i[y_i]$.

We wish to analyze the impact of the VIB criteria on the mutual information lower bound derived in Proposition 1. To establish a relationship between the original time series chunk $S_{i,k}^j$ and the VIB influenced latent space representation $z_{i,k}^{(j)}$, we consider $\hat{S}_{i,k}^j = f(z_{i,k}^{(j)})$ and $p_{i,k}^{(j)} = \mathbb{P}[\hat{S}_{i,k}^j \neq S_{i,k}^j]$ to be the predicted token for sampled embedding $z_{i,k}^{(j)}$ and the associated token-level prediction error respectively. Thus, we present Proposition 2 to characterize the relationship between $S_{i,k}^j$ and $z_{i,k}^{(j)}$.

**Proposition 2.** *Given $|\mathcal{V}| \geq 2$, for $p_e^{(j,i,k)} \in \big( 0, 1 - \frac{1}{|\mathcal{V}|} \big]$, the upper bound of token prediction error $p_e^{(j,i,k)}$ is inversely related to mutual information between token $S_{i,k}^j$ and its latent embedding $z_{i,k}^{(j)}$.*

*Proof.* Proof provided in Appendix B.2 □

Proposition 2 shows that as the token-wise error $p_e^{(j,i,k)}$ decreases, the mutual information $I(S_{i,k}^j; z_{i,k}^{(j)})$ admits a higher lower bound, suggesting that better token prediction corresponds to more informative embeddings.

Next, we are interested in characterizing the mutual information between the label $y_i \in \{1, \ldots C\}$, and the low dimensional embedding sequence $Z_i = \{Z_{i,1}, Z_{i,2} \ldots Z_{i,K}\}$, where $Z_{i,k} = \{z_{i,k}^{(1)}, z_{i,k}^{(2)}, \ldots z_{i,k}^{(M)}\}$. Therefore, we provide a lower bound between labels $y_i$ and the embedding sequence $Z_i$ in Proposition 3.

**Proposition 3.** *The lower bound of mutual information between label $y_i \in \{1, \ldots C\}$ and stochastic embedding $Z_i$ depends on entropy of labels and the negative log likelihood loss, and is given by:*

$$I(y_i; Z_i) \geq \mathbb{H}(y_i) - \mathcal{L}_{\text{NLL}}.$$

*Proof.* Proof provided in Appendix B.3 □

We observe that the lower bound in Proposition 3 depends solely on the NLL term. However, in VIB training, the total loss also includes the KL divergence term as in equation 22. The KL component acts as a secondary objective that encourages compression of $Z_i$. Using Proposition 3, we can make critical observations regarding the role of $\beta$ in determining the trade-off between predictive performance and compression. With a smaller value of $\beta$, the loss is dominated by the NLL term, making the model focus on predictive accuracy and increasing the lower bound on $I(y_i, Z_i)$. However, weak regularization can lead to overfitting in such scenarios as well.

On the other hand, with a large $\beta$, the KL term dominates leading to a higher emphasis on compression of embedding $Z_i$ leading to discarding input information to enforce a tighter bottleneck. In this case, excessive compression can reduce the lower bound on $I(y_i; Z_i)$ if the NLL term rises due to lost predictive information. Together with Proposition 2, the insights resulting from Proposition 3 highlight that higher mutual information in $Z$ simultaneously suppresses token errors and preserves label information. We will finalize this section with a discussion on best practices for implementation based on the structural properties of the bounds.

### 3.5 The ADEPT Framework

We define two versions of the *ADEPT* framework. ADEPT v1.0 is the baseline version of our pipeline, which directly applies pretrained text embeddings to serialized time-series segments, followed by a multi-head attention classifier. ADEPT v2.0 extends this baseline by incorporating a VIB layer, which compresses the raw embeddings into compact, task-relevant codes that reduce noise and improve generalization. While both versions eliminate the need for traditional data engineering steps, ADEPT v2.0 introduces an additional mechanism to better align learned representations with downstream prediction objectives. The general pipeline of ADEPT v2.0 is shown in Figure 2 and Algorithm 1.

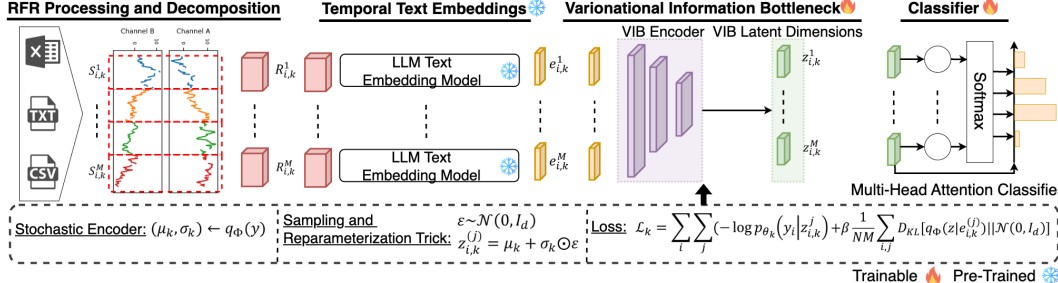

Figure 2: Illustration of the ADEPT v2.0. Framework

## 4 Experimental Results

This section evaluates the predictive performance of the proposed ADEPT framework across datasets drawn from diverse domains, including healthcare, science, finance, and IoT. These datasets introduce a large variety of challenges such as data integrity, temporal dependencies, and privacy constraints. The prediction tasks also vary significantly in complexity—from ternary classification of Bitcoin price direction to multi-class root cause analysis in hydroelectric systems, where each failure mode is represented by only a handful of observations. To tackle these challenges, existing approaches rely on heavily customized data engineering pipelines, manually optimized for each dataset and task. In our experiments, we benchmark ADEPT against these application-specific state-of-the-art models. Details on datasets, evaluation metrics, and implementation are provided in Appendices D, E, and F. In addition to the results here, an ablation study is also provided in Appendix C.

### 4.1 Science Application - *Predicting the Astrophysical Class of Light Curves*

Table 1a compares the results from our two frameworks—*ADEPT v1.0* (without VIB) and *ADEPT v2.0* (with VIB)—against three state-of-the-art classifiers that yield the best accuracies in literature: CATS Fraga et al. (2024), AMPEL Nordin et al. (2025), and ORACLE Shah et al. (2025). The benchmark models employ conventional time-series pipelines with multiple preprocessing steps; and

---

**Algorithm 1** ADEPT v2.0

---

**Require:** Series $S_{i,k}$, embedding model $g$, VIB params $\phi_k, \theta_k$, Transformer-AE params $\psi$, hyperparams $(\beta, \eta, \eta', \eta'', E_{\text{VIB}}, E_{\text{AE}}, E_{\text{CL}})$

1: Partition $S_{i,k}$ into $M$ segments of length $L = \frac{T}{M}, \forall i \in [N], \forall k \in [K]$.      ▷ RFR Decomposition

2: Compute embeddings $\mathbf{e}_{i,k}^{(j)}$ (Eq. 3) using text embedding model $g$.      ▷ Temporal Text Embeddings

3: **for** $k = 1, \ldots, K$ **do**      ▷ Variational Information Bottleneck

4:      **for** epoch $= 1, \ldots, E_{\text{VIB}}$ **do**

5:          Compute the per-view VIB loss $\mathcal{L}_k$ using (Eqs. 9) and update $\phi_k, \theta_k$ via Adam($\eta$).

6:      **end for**

7:      Utilize updated $\phi_k, \theta_k$ to obtain latent space encodings $\{\mathbf{z}_{i,k}^{(j)}\}$.

8: **end for**

9: **for** epoch $= 1, \ldots, E_{\text{AE}}$ **do**      ▷ Transformer-AE pretraining

10:      Assemble each view's latent encodings, compute loss $\mathcal{L}_{AE}$ (Eq. 10) and update $\psi$ via Adam($\eta'$).

11: **end for**

12: **for** epoch $= 1, \ldots, E_{\text{CL}}$ **do**      ▷ Classification fine-tuning

13:      Compute per-view class probabilities $p_{i,k} = \text{Softmax}\big(T_\psi(Z_{i,k})\big), \forall k \in K$.

14:      Fuse the per-view probabilities into a final distribution $p_i$ (Eq. 11).

15:      Compute negative log-likelihood loss $\mathcal{L}_{NLL} = \frac{-1}{N} \sum_{i=1}^{N} \log p_i[y_i]$ and update $\psi$ via Adam($\eta''$).

16: **end for**

17: Use Transformer-AE $\psi$ to obtain fused probabilities $p_i$ for predicting $\hat{y}_i = \arg\max_c p_i[c]$      ▷ Inference

---

leverage specialized architectures like CNN+LSTM hybrids or hierarchical RNNs to extract temporal features from the multi-band photometric data. Their reported classification accuracies range from 80% to 84%. Unlike these methods — which rely on hand-crafted time and color features followed by gradient-boosted trees or hierarchical RNNs — our pipelines operate directly on text-serialized light curves. *ADEPT v1.0* achieves 95.98 % accuracy, while incorporation of VIB in *ADEPT v2.0* further improves accuracy to 97.83 %, outperforming all benchmarks with an improvement of >10%.

### 4.2 HEALTHCARE APPLICATION - *Predicting Patient Condition using EEG Data*

Prediction results are shown in Table 1b. For this dataset, we use three high-performing benchmarks from recent literature: MiniRocket Keshavarzian et al. (2023), MHCAN Huang et al. (2024), and TSEM Pham et al. (2023). These methods follow conventional multivariate time-series classification pipelines, using application-specific preprocessing steps such as wavelet decomposition, temporal convolutions, and spatiotemporal mapping, coupled with specialized architectures like transformers and hybrid CNN-RNN models. Their reported classification accuracies range from 59.0% to 75.60%, which constitutes a significant spread, showcasing that the capability of inherent indicators of mental state are challenging to discover. Among the proposed models, *ADEPT v1.0*, despite its simplicity, achieves a comparable 58.97% accuracy. With the addition of a variational information bottleneck in *ADEPT v2.0*, accuracy improves to 73.68%, outperforming two of the three benchmarks and closely approaching the best-performing method. An interesting observation in this experiment is the significant accuracy gap between *ADEPT v1.0* and *v2.0* models. The VIB step in *ADEPT v2.0* unlocks a richer representation of data and results in 14.7% improvement in accuracy.

### 4.3 FINANCE APPLICATION - *Predicting Future Bitcoin Price Trend*

Table 1c compares our framework, *ADEPT v1.0* and *ADEPT v2.0*, against three baseline methods from the literature: a Recurrent LSTM Kwon et al. (2019), an Ensemble Deep Learning Rao et al. (2023), and a BiLSTM Critien et al. (2022). These baselines rely on traditional financial preprocessing steps including normalization, anomaly removal, and hand-engineered feature selection, with accuracies ranging from 64% to 77.20%. In this application, *ADEPT v1.0* achieves a poor accuracy of 45.40%. However, incorporating VIB in *ADEPT v2.0* significantly boosts performance to 88.49%. This is the best-performing model across benchmarks that have access to the same data.

### 4.4 IoT APPLICATION - *Predicting the Cause of Hydropower Reliability Issues*

The IoT application data is proprietary, thus, there is no prior work on prediction in this dataset. We take this opportunity to test the performance of the existing AutoML methods, specifically the TSFEL package Barandas et al. (2020). An additional challenge comes from the privacy and data residency requirements for this dataset preventing the use of public LLM-based embedding models such as OpenAI's `text-embedding-3-small`. Therefore, we leverage the `nomic-embed-text-v1`

Table 1: Benchmarking Results Across a Range of Applications

(a) Science Dataset: *PLAsTiCC Classification*

| Source | Preprocessing Steps | Classifier | Accuracy |
|---|---|---|---|
| Fraga et al. (2024) | *Clean data, filter, normalize, etc.* | CNN+LSTM | 83% |
| Nordin et al. (2025) | *Filter, remove noise, negative-flux, etc.* | ParSNIP+GBM | 80% |
| Shah et al. (2025) | *Remove noise, truncate, pad, mask, etc.* | RNN | 84% |
| *ADEPT v1.0* | *Bypassed via text embedding* | MHA | 95.98% |
| ***ADEPT v2.0*** | ***Bypassed via text embedding + VIB*** | **MHA** | **97.83%** |

(b) Healthcare Dataset: *SelfRegulationSCP2 Classification*

| Source | Preprocessing Steps | Classifier | Accuracy |
|---|---|---|---|
| Keshavarzian et al. (2023) | *Augment via random freq. butchering, etc.* | MiniRocket | 59.0% |
| Huang et al. (2024) | *1D-Conv, positional encoding, etc.* | MHCA | 62.20% |
| Pham et al. (2023) | *2D-Conv filters, spatiotemporal maps, etc.* | Transformer | 75.60% |
| *ADEPT v1.0* | *Bypassed via text embedding* | MHA | 58.97% |
| ***ADEPT v2.0*** | ***Bypassed via text embedding + VIB*** | **MHA** | **73.68%** |

(c) Financial Dataset: *Bitcoin Price Trend Classification*

| Source | Preprocessing Steps | Classifier | Accuracy |
|---|---|---|---|
| Kwon et al. (2019) | *Forward-fill missing data, drop outliers, etc.* | LSTM | 66% |
| Rao et al. (2023) | *Remove anomaly, sequence structure, etc.* | CNN–LSTM | 64% |
| Critien et al. (2022) | *Reduce noise, compute sentiment, etc.* | BiLSTM | 77.20% |
| *ADEPT v1.0* | *Bypassed via text embedding* | MHA | 45.40% |
| ***ADEPT v2.0*** | ***Bypassed via text embedding + VIB*** | **MHA** | **88.49%** |

(d) Internet-of-Things Dataset: *Hydropower-Research Institute Fault Classification*

| Model | Preprocessing Steps | Accuracy | Top-2 Accuracy |
|---|---|---|---|
| *TSFEL+MHAN* | *TSFEL features, MI selection, normalize, etc.* | 42.80% | **57.14%** |
| *ADEPT v1.0* | *Bypassed via text embedding* | 45.00% | **66.67%** |
| ***ADEPT v2.0*** | ***Bypassed via text embedding + VIB*** | **74.35%** | **97.5%** |

model (768-dim) hosted on-prem to embed each text segment. Details of the implementation for this dataset is provided in Appendix F. Table 1d reports overall test accuracy: Feature extraction + MHAN achieves 42.80%, ADEPT v1.0 45.00%, and ADEPT v2.0 attains 74.35%. This substantial gain showcases the advantage of combining semantic embeddings with VIB filtering.

In many IoT-enabled asset monitoring applications, analyzing the performance of the model's second-best prediction—referred to as Top-2 Accuracy—can be particularly valuable. Top-2 predictions offer actionable insights by identifying plausible alternative failure modes, which can guide proactive inspections, and trigger early intervention to prevent failures of heavy assets like turbines or thrust bearings in hydropower systems, potentially saving millions of dollars per incident. On this metric, *ADEPT v2.0* has an accuracy of 97.5%, which offers a significant improvements over the benchmarks.

## 5 CONCLUSION

We have shown that general-purpose text embeddings—without any additional feature engineering or domain-specific data preprocessing—can serve as powerful representations for raw time-series classification. Across four diverse experiments (Science, Healthcare, Finance and IoT), ADEPT consistently outperforms application specific engineered predictive models. We demonstrate that off-the-shelf text embedding models, when paired with a lightweight variational information bottleneck step, can capture the salient structure of heterogeneous time-series inputs. This paves the way for fast, turnkey classification solutions in domains where feature engineering is costly or impractical. Experiments across diverse datasets show that the *ADEPT v2.0* model consistently matches or surpasses the best-performing benchmarks in all application domains; showcasing that text embeddings supplemented with VIB can perform the function of cost-effective and capable data engineers.

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

# A ADEPT v2.0 ALGORITHM

The full pseudoscope of ADEPT v2.0 is discussed in Algorithm 2

---

**Algorithm 2** ADEPT v2.0

---

**Require:** Multivariate series $S_{i,k} \in \mathbb{R}^{T \times F}$, chunks $M$, frozen text embedder $g$, VIB params $\phi_k, \theta_k$, Transformer-AE $T_\psi$, hyperparams $(\beta, \eta, B, E_{\text{VIB}}, E_{\text{AE}}, E_{\text{CL}})$
**Ensure:** Predicted label $\hat{y}_i$

1: **RFR Processing and Decomposition**
2: $L \leftarrow T/M$
3: **for** $k = 1, \ldots, K$ **do**
4:      **for** $j = 1, \ldots, M$ **do**
5:          $S_{i,k}^{(j)} \leftarrow S_{i,k}[(j-1)L + 1 : jL, \, 1 : F]$
6:      **end for**
7: **end for**
8: **Temporal Text Embeddings**
9: **for** $k = 1, \ldots, K$ **do**
10:      **for** $j = 1, \ldots, M$ **do**
11:          $R_{i,k}^{(j)} = \text{serialize}(S_{i,k}^{(j)})$
12:          $\mathbf{e}_{i,k}^{(j)} = g\big(R_{i,k}^{(j)}\big) \in \mathbb{R}^E$                    ▷ (Eq. 3)
13:      **end for**
14: **end for**
15: **Variational Information Bottleneck**
16: **for** $k = 1, \ldots, K$ **do**
17:      **for** $j = 1, \ldots, M$ **do**
18:          $\mu_k = W_\mu^{(k)} \mathbf{e}_{i,k}^{(j)} + b_\mu^{(k)}, \quad \log \sigma_k^2 = W_{\log \varphi}^{(k)} \mathbf{e}_{i,k}^{(j)} + b_{\log \varphi}^{(k)}$      ▷ (Eq. 5)
19:          $\sigma_k = \exp\big(\tfrac{1}{2} \log \sigma_k^2\big)$
20:          Sample $\varepsilon \sim \mathcal{N}(0, I_d)$
21:          $\mathbf{z}_{i,k}^{(j)} = \mu_k + \sigma_k \odot \varepsilon$                       ▷ (Eqs. 6, 7)
22:          $\ell_{i,k}^{(j)} = W_y^{(k)} \mathbf{z}_{i,k}^{(j)} + b_y^{(k)}, \quad p_{\theta_k}(y_i \,|\, \mathbf{z}) = \text{Softmax}(\ell_{i,k}^{(j)})$      ▷ (Eq. 8)
23:      **end for**
24:      Compute per-view loss

$$\mathcal{L}_k = \frac{1}{NM} \sum_{i,j} \big[ -\log p_{\theta_k}(y_i \mid z) \big] + \beta \frac{1}{NM} \sum_{i,j} D_{KL}\big[ q_{\phi_k}(\cdot) \| \mathcal{N}(0, I) \big]$$

▷ (Eq. 9)

25:      Update $\phi_k, \theta_k$ via Adam($\eta$) for $E_{\text{VIB}}$ epochs
26: **end for**
27: **Transformer-based Classifier Design**
28: Assemble each view's sequence $Z_{i,k} = [\, \mathbf{z}_{i,k}^{(1)}, \ldots, \mathbf{z}_{i,k}^{(M)} \,]$
29: *(a) Pre-trained autoencoder:*

$$\mathcal{L}_{AE} = \frac{1}{KN} \sum_{i,k} \frac{1}{M \, 2E} \big\| \hat{Z}_{i,k} - Z_{i,k} \big\|_Z^2 \quad \text{(Eq. 10)}$$

30: Update $\psi$ via Adam($\eta\prime$) for $E_{\text{AE}}$ epochs
31: *(b) Fine-tune for classification:*
32: **for** $k = 1, \ldots, K$ **do**
33:      $p_{i,k} = \text{Softmax}\big( T_\psi(Z_{i,k}) \big)$
34: **end for**
35: Fuse

$$p_i = \frac{p_{i,1} \odot \cdots \odot p_{i,K}}{\sum_c [\, p_{i,1} \odot \cdots \odot p_{i,K} \,]_c} \quad \text{(Eq. 11)}$$

36: Minimize

$$\mathcal{L}_{NLL} = -\frac{1}{N} \sum_i \log p_i[y_i]$$

37: Update $\psi$ via Adam($\eta\prime\prime$) for $E_{\text{CL}}$ epochs
38: **return** $\hat{y}_i = \arg\max_c p_i[c]$

---

# B THEORETICAL PROOFS

## B.1 PROOF OF PROPOSITION 1

*Proof.* Using the chain rule of mutual information as denoted by equation 12 provides a valid lower bound under the assumption of approximate independence between tokens (commonly assumed in serialized inputs).

$$I(S_{i,k}^j; e_{i,k}^{(j)}) = \sum_{l=1}^{n} I(S_{l,k}^j; E \mid S_{<l,k}^j) \geq \sum_{l=1}^{n} I(S_{l,k}^j; E) \tag{12}$$

Now, we let $\hat{S}_{i,k}^j = f(E)$ be a decoder predicting token $S_{i,k}^j$ from embedding $E$, and define the token-wise prediction error:

$$p_e^{(j,i,k)} = \mathbb{P}[\hat{S}_{i,k}^j \neq S_{i,k}^j] \tag{13}$$

Using Fano's inequality Cover (1999) yields equation 14

$$\mathbb{H}(S_{i,k}^j \mid E) \leq H_b(p_e^{(j,i,k)}) + p_e^{(j,i,k)} \log(|\mathcal{V}| - 1) \tag{14}$$

In equation 14, $H_b(p) = -p \log p - (1-p) \log(1-p)$ is the binary entropy function, and $|\mathcal{V}|$ is the number of classes. Next, the definition of mutual information can be provided based on equation 15

$$I(S_{i,k}^j; E) = \mathbb{H}(S_{i,k}^j) - \mathbb{H}(S_{i,k}^j \mid E) \tag{15}$$

Combining equation 14 and equation 15, we get

$$I(S_{i,k}^j; E) \geq \mathbb{H}(S_{i,k}^j) - H_b(p_e^{(j,i,k)}) - p_e^{(j,i,k)} \log(|\mathcal{V}| - 1) \tag{16}$$

Summing equation 16 over all tokens and using the relationship in equation 12, we can conclude

$$I(S_{i,k}^j; e_{i,k}^{(j)}) \geq \sum_{l=1}^{n} I(S_{l,k}^j; E) \geq \sum_{l=1}^{n} \left[ \mathbb{H}(S_{l,k}^j) - H_b(p_e^{(j,l,k)}) - p_e^{(j,l,k)} \log(|\mathcal{V}| - 1) \right] \tag{17}$$

$\square$

## B.2 PROOF OF PROPOSITION 2

*Proof.* We know from Proposition equation 1 that

$$I(S_{i,k}^j; Z) \geq \mathbb{H}(S_{i,k}^j) - H_b(p_e^{(j,i,k)}) - p_e^{(j,i,k)} \log(|\mathcal{V}| - 1), \tag{18}$$

We consider the function $f(p)$, defined in equation 19 where $H_b(p) = -p \log p - (1-p) \log(1-p)$ represents binary cross entropy.

$$f(p) = H_b(p) + p \log(|\mathcal{V}| - 1) \tag{19}$$

We note from equation 19 that $f(p)$ is strictly increasing on the interval $p \in (0, 1 - \frac{1}{|\mathcal{V}|}]$ (the regime of interest where the classifier performs better than random guessing), its inverse $f^{-1}$ exists and is strictly increasing on the corresponding range. Therefore rearranging Fano's bound leads us to equation 20.

$$p_e^{(j,i,k)} \leq f^{-1}\left(\mathbb{H}(S_{i,k}^j) - I(S_{i,k}^j; Z)\right), \tag{20}$$

Next, we examine the derivative of $f(p)$ as given in equation 21

$$f'(p) = \log \frac{(1-p)(|\mathcal{V}| - 1)}{p}. \tag{21}$$

We can see that $f(p)$ is strictly increasing on $(0, 1 - \frac{1}{|\mathcal{V}|}]$. Since, we assume that $|\mathcal{V}| \geq 2$, we can say that the derivative is positive and as a result $f^{-1}$ is well-defined and monotonic.

The monotonic, strictly increasing nature of $f^{-1}$ established in equation 21 and equation 20 imply that the upper bound of token-prediction error has an inverse relationship with the mutual information between token $S_{i,k}^j$ and the latent embedding $Z$.

$\square$

### B.3 PROOF OF PROPOSITION 3

*Proof.* We know that the VIB training objective is given by equation 22

$$\mathcal{L}_{\mathrm{VIB}} = \underbrace{\mathbb{E}[-\log p_\theta(y_i \mid Z_i)]}_{\mathcal{L}_{\mathrm{NLL}}} + \beta \underbrace{\mathbb{E}[D_{\mathrm{KL}}[q_{\phi_k}(Z_i \mid e_{i,k}^{(j)}\|\mathcal{N}(0, I_d)]]}_{\mathcal{L}_{\mathrm{KL}}}, \qquad (22)$$

In equation 22, $\mathcal{L}_{\mathrm{NLL}}$ is the negative log-likelihood (cross-entropy loss), $\mathcal{L}_{\mathrm{KL}}$ is the KL regularization term and $\beta \geq 0$ controls the prediction vs. compression trade-off.

Further, the mutual information between the label $y_i$ and the stochastic embedding $Z_i$ is given by equation 23

$$I(y_i; Z_i) = \mathbb{H}(y_i) - \mathbb{H}(y_i \mid Z_i). \qquad (23)$$

Using the variational decoder, the conditional entropy is upper-bounded by the NLL:

$$\mathbb{H}(y_i \mid Z_i) \leq \mathbb{E}[-\log p_\theta(y_i \mid Z_i)] = \mathcal{L}_{\mathrm{NLL}}. \qquad (24)$$

This gives the lower bound denoted by equation 25

$$I(y_i; Z_i) \geq \mathbb{H}(y_i) - \mathcal{L}_{\mathrm{NLL}}. \qquad (25)$$

$\square$

## C ABLATION STUDY: EMBEDDING QUALITY AND VIB SENSITIVITY

To better understand the performance behavior of ADEPT, we conducted an ablation study on the Healthcare dataset (SelfRegulationSCP2). This study focuses on two aspects: (i) the impact of embedding model quality, and (ii) the sensitivity of ADEPT to the VIB bottleneck dimension.

We evaluated ADEPT v2.0. using three text embedding models of increasing capacity—nomic ($d = 768$), ada ($d = 1536$), and large ($d = 3072$)—while varying the VIB bottleneck dimension $(64, 128, 256, 512)$. For reference, we also report the baseline ADEPT v1.0. without VIB and without text embedding model (No-TE: just a classification head).

Table 2: Performance of ADEPT Variants with Different VIB Dimensions on the Healthcare Dataset.

| Model | VIB Dim. | No-TE | nomic | ada-002 | 3-large |
|---|---|---|---|---|---|
| ADEPT v1.0. | – | 51.32% | 58.5% | 63.2% | 66.7% |
| ADEPT v2.0. | 64 | – | 73.7% | 70.1% | 69.8% |
| ADEPT v2.0. | 128 | – | **81.6%** | **75.3%** | 70.6% |
| ADEPT v2.0. | 256 | – | 73.7% | 68.4% | **77.9%** |
| ADEPT v2.0. | 512 | – | – | – | 65.7% |

The results in Table 2 provide several key insights (Bold numbers indicate the best performance for each embedding model.):

1. *Impact of embedding quality.* Higher-dimensional embeddings (e.g., large, $d = 3072$) provide stronger baselines, confirming that embedding richness improves performance even without VIB.

2. *Optimal VIB bottleneck.* Moderate VIB dimensions ($d_{\mathrm{VIB}} = 128$) consistently yield the best trade-off between preserving informative features and preventing overfitting. Extremely small or large bottlenecks reduce accuracy due to either under-representation or excessive compression.

3. *Stronger VIB gains in low-dimensional embeddings.* The relative improvement from VIB is most pronounced for the nomic embedding ($+23.1\%$ absolute gain from 58.49% to 81.58%). This occurs because low-dimensional embeddings initially capture fewer task-specific features; the VIB bottleneck acts as a targeted information filter, forcing the model to retain only the most predictive signals and discard noisy dimensions.

4. *Why nomic achieves a better final accuracy.* While all the embedding models showcase comparable accuracy for this particular case, we observe that nomic achieves the highest performance.

Interestingly, the best overall accuracy is obtained with the `nomic` embedding ($d = 768$) combined with VIB ($d_{\text{VIB}} = 128$). Smaller embeddings are inherently less redundant, so the VIB module primarily performs *denoising* rather than aggressive compression. In contrast, larger embeddings encode many irrelevant or domain-misaligned features for this healthcare dataset, making them more prone to overfitting after bottleneck compression. Consequently, the synergy between compact embeddings and VIB leads to superior generalization for `nomic`.

Overall, these results indicate that ADEPT's improvements arise from a combination of high-quality embeddings and the VIB-enhanced architecture.

## D  DATASETS AND PREDICTION TASKS

### D.1  SCIENCE APPLICATION - *PLAsTiCC Dataset*

The science application experiment is from an astrophysics example (PLAsTiCC dataset from the *2018* Kaggle competition) that aims to predict the astrophysical class of light curves. The dataset consists of approximately $7,848$ simulated LSST light curves in six filters $(u, g, r, i, z, y)$ and labels for 14 astrophysical classes (e.g., Type Ia/II supernovae, RR Lyrae). Curve lengths vary from 50 to 350 epochs, and because observations only occur when each field is visible (weather, scheduling, maintenance), roughly 30 % of per-band flux measurements are missing on average.

### D.2  HEALTHCARE APPLICATION - *SelfRegulationSCP2 Dataset*

The healthcare application focuses on classification using the SelfRegulationSCP2 dataset, a multivariate time-series dataset derived from electroencephalography (EEG) recordings. Each record comprises eight scalp channels sampled at 250 Hz. During each session, subjects receive a visual cue and then attempt either to increase ("up") or decrease ("down") their brain signal over a 5 s interval, preceded by a 2 s baseline. There are 200 trials per subject—100 "up" and 100 "down"—resulting in a balanced, two-class (binary) classification task. This clean, well-labeled dataset is ideal for evaluating and comparing EEG-based decoding methods.

### D.3  FINANCE APPLICATION - *Bitcoin Price Trend Dataset*

The financial application focuses on next-day trend classification using the Bitcoin Price Trend Dataset. The task is framed as a 3-class classification problem—predicting whether the price will rise by more than 1%, fall by more than 1%, or remain stable within 1%. Spanning daily BTC/USD data from 2015–2023, each record includes OHLCV (open, high, low, close, volume) plus 14 technical indicators: RSI-7, RSI-14, CCI-7, CCI-14, SMA-50, EMA-50, SMA-100, EMA-100, MACD, Bollinger Bands, True Range, ATR-7, and ATR-14. Our target is next-day price movement—classified as *positive* ($> +1$ %), *negative* ($< -1$ %), or *stable* ($|\%| \leq 1\%$). We train and tune on data through 2015 to 2022, and evaluate on 2023 (365 days), yielding an approximate class balance of 34 % positive, 27 % negative, and 38 % stable.

### D.4  HYDROPOWER RESEARCH INSTITUTE (HRI) DATASET

IoT application focuses on predicting the root cause of reliability issues in hydropower components using a proprietary commercial dataset shared with the authors through the courtesy of the Hydropower Research Institute. The original dataset encompasses information from 197 hydropower plants and 844 generating units, which accounts for approximately 42% of U.S. capacity. The data includes operational metrics, and event logs. We construct the training dataset by aggregating a set of reliability events into a database that pairs each event's cause code with multi-stream sensor readings captured over several days leading up to the event. There are 14 unique cause codes. The objective is to predict the right cause code subject to the inherent variability of industrial data due to highly dynamic and heterogeneous conditions, which introduces substantial complexity.

Although the raw feed is nominally logged every 30 s, individual channels (91 channels) report at irregular intervals (some hourly, others daily or weekly), leaving substantial gaps. Maintenance logs record the exact timestamp of each failure along with one of 14 high-level cause codes (e.g., Main transformer, Shaft packing, Transmission line). We therefore set $k = 3$ views for each event: we extract three contiguous 6 h windows of sensor readings immediately preceding the failure—covering 0–6 h, 6–12 h, and 12–18 h before the event—and assign the corresponding cause code as the label.

In total, this yields 390 events (each with 3 windows), which we split chronologically into 80% for training, 10% for validation, and 10% for testing.

## E    Evaluation Metrics

We report overall accuracy, per-class precision, recall, and $F_1$-scores (both micro- and macro-averaged) to quantify classification performance. To illustrate the impact of applying variational information bottleneck on representation quality, we visualize raw text embeddings and IB-filtered embeddings using t-SNE plots, showing improved cluster separation. We also include normalized confusion matrices to highlight class-wise true and false positive rates. All metrics and visualizations are computed on held-out test splits for each dataset, ensuring a consistent and robust assessment of our model.

## F    Implementation Details

*PLAsTiCC:* Each light curve is divided into $M = 10$ equal-duration segments and embedded via the OpenAI Text Embedding (text-embedding-3-small, 1536 dim) model. VIB filter is trained with d $= 256$, epochs $= 100$, batch size $= 4$, lr $= 1 \times 10^{-4}$, and $\beta = 1 \times 10^{-4}$. The Transformer classifier uses $h = 128$, $n_{\text{head}} = 32$, $L = 2$, and dim$_{\text{ff}} = 128$. The Transformer classifier settings are identical (two layers, $n_{\text{head}} = 32$, dim$_{\text{ff}} = 128$), with autoencoder and clustering pretraining for 100 and 50 epochs, respectively.

*SelfRegulationSCP2 :* Each EEG trial is divided into $M = 24$ equal-duration segments and embedded via the OpenAI Text Embedding (text-embedding-3-small, 1536-dim) model. VIB filter is trained with d $= 256$, epochs $= 100$, batch size $= 4$, lr $= 1 \times 10^{-4}$, and $\beta = 1 \times 10^{-4}$. The Transformer classifier uses $h = 128$, $n_{\text{head}} = 32$, $L = 2$, and dim$_{\text{ff}} = 128$, with autoencoder and clustering pretraining for 50 and 50 epochs, respectively.

*Bitcoin Price Trend:* We take the most recent 15 days of data per sample, segmented into $M = 5$ non-overlapping 3-day windows, and embed each window using the OpenAI Text Embedding (small, 1536 dim) model. The VIB uses the same hyperparameters as above. The Transformer classifier employs $h = 128$, $n_{\text{head}} = 16$, $L = 2$, and dim$_{\text{ff}} = 128$. Autoencoder pretraining runs for 100 epochs and clustering pretraining for 200 epochs.

*HRI:* This is a commercial dataset, so we embed using the nomic-embed-text-v1 (765 dim). We extract three consecutive 6-hour windows immediately preceding each failure $k = 3$, each split into $M = 24$ non-overlapping chunks. The VIB is trained with d $= 256$, epochs $= 100$, batch size $= 4$, lr $= 1 \times 10^{-4}$, and $\beta = 1 \times 10^{-4}$. The Transformer classifier uses $h = 128$, $n_{\text{head}} = 32$, $L = 2$, and dim$_{\text{ff}} = 128$. We pretrain the autoencoder for 100 epochs and the clustering head for 50 epochs.

To evaluate the effectiveness of our proposed ADEPT pipeline on the proprietary HRI dataset—and in the absence of any publicly available benchmark—we instantiate and compare three classification strategies:

1. *Feature extraction + Classifier*: We linearly interpolate missing readings onto a uniform $30\,\text{s}$ grid, slide 15 min windows over each event, extract over 9,000 time- and frequency-domain features per channel via TSFEL, select the top 100 via mutual information, normalize, and classify with same MHA classifier as the ADEPT framework has.

2. *ADEPT v1.0*: We serialize each 15 min segment and embed it offline to a 768-dim vector via the `nomic-embed-text-v1` model, then classify directly.

3. *ADEPT v2.0*: Our full pipeline, where VIB compresses the 768-dim embeddings before fusion and classification.

## G    Detailed Results on Predicting the Astrophysical Class of Light Curves

Figure 3 presents a 3D t-SNE projection of 1536-dimensional segment embeddings from the PLAsTiCC-2018 LSST dataset, colored by transient class (14 astrophysical types). *Left:* raw OpenAI text embeddings exhibit overlapping and diffuse clusters. *Right:* embeddings after Variational Information Bottleneck (VIB) filtering show tighter, well-separated clusters, demonstrating the effectiveness of VIB in ADEPT v2.0.

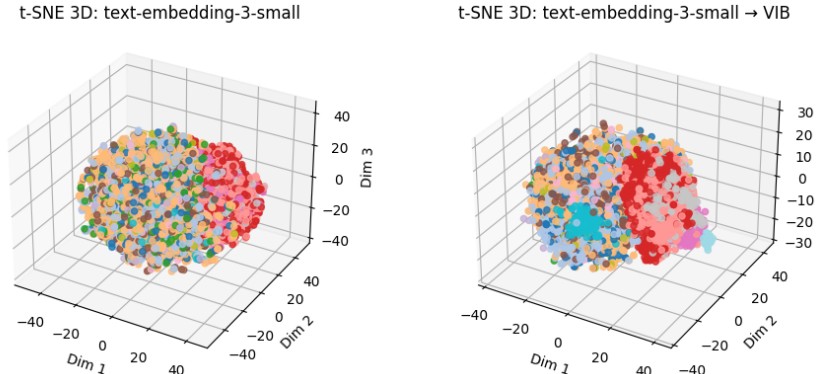

Figure 3: 3D t-SNE projection of 1536-dim segment embeddings from the PLAsTiCC-2018 LSST dataset, colored by transient class. *Left:* raw text embeddings; *Right:* embeddings after VIB filtering.

Table 3 reports per–class precision, recall, and $F_1$–scores for the 14 astrophysical classes in the PLAsTiCC-2018 test set, alongside overall accuracy (0.98), macro– and weighted–average metrics. Nearly perfect scores are achieved on most classes, with minor drops for classes 42 and 67. Figure 4

Table 3: Per–class performance on PLAsTiCC

| Class | Precision | Recall | $F_1$–score |
|---|---|---|---|
| 6 | 1.00 | 1.00 | 1.00 |
| 15 | 1.00 | 0.98 | 0.99 |
| 16 | 1.00 | 1.00 | 1.00 |
| 42 | 0.92 | 0.97 | 0.94 |
| 52 | 0.85 | 0.97 | 0.91 |
| 53 | 1.00 | 1.00 | 1.00 |
| 62 | 0.94 | 0.91 | 0.92 |
| 64 | 1.00 | 1.00 | 1.00 |
| 65 | 1.00 | 1.00 | 1.00 |
| 67 | 0.94 | 0.81 | 0.87 |
| 88 | 1.00 | 0.97 | 0.99 |
| 90 | 1.00 | 0.99 | 1.00 |
| 92 | 1.00 | 1.00 | 1.00 |
| 95 | 1.00 | 1.00 | 1.00 |
| **Accuracy** | — | — | 0.98 |
| **Macro avg** | 0.98 | 0.97 | 0.97 |
| **Weighted avg** | 0.98 | 0.98 | 0.98 |

shows the normalized confusion matrix for our VIB embedding pipeline on PLAsTiCC: rows correspond to true classes and columns to predicted classes; cell intensity indicates per-class recall. Misclassifications are rare and primarily occur among classes with similar light-curve signatures.

# H   DETAILED RESULTS ON PREDICTING PATIENT CONDITION USING EEG DATA

Figure 5 presents a 3D t-SNE projection of 1536-dimensional segment embeddings for the SelfRegulationSCP2 dataset, colored by class. *Left:* raw OpenAI text-embedding-3-small embeddings exhibit diffuse, overlapping clusters. *Right:* embeddings after Variational Information Bottleneck (VIB) filtering form tighter, more separable clusters.

Table 4 reports per–class precision, recall, and $F_1$–scores for the two sentiment classes in our test set, alongside overall accuracy (0.74), macro– and weighted–average metrics. Performance is balanced across classes, with "negativity" achieving 0.76 on all metrics and "positivity" slightly lower at 0.71. Figure 8 shows the normalized confusion matrix for the ADEPT v2.0 pipeline on the SelfRegulationSCP2 dataset. Rows correspond to true movement classes (negativity, positivity) and columns to predicted classes; cell intensity indicates per–class recall.

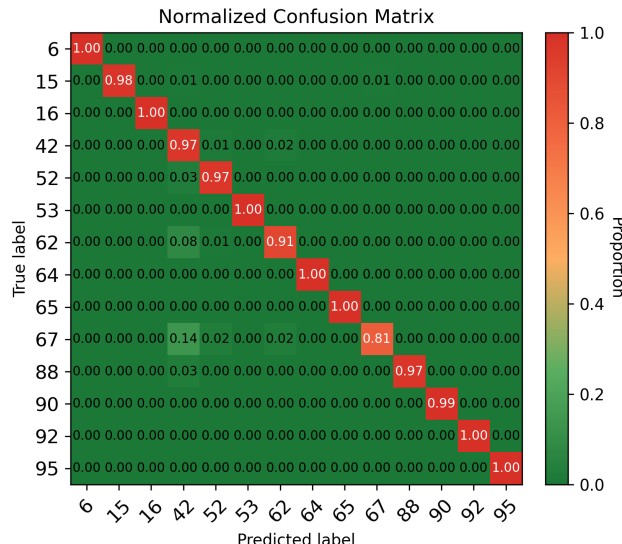

Figure 4: Normalized confusion matrix for the IB-filtered pipeline on the PLAsTiCC-2018 LSST dataset. Rows correspond to true classes and columns to predicted classes; cell intensity indicates per-class recall.

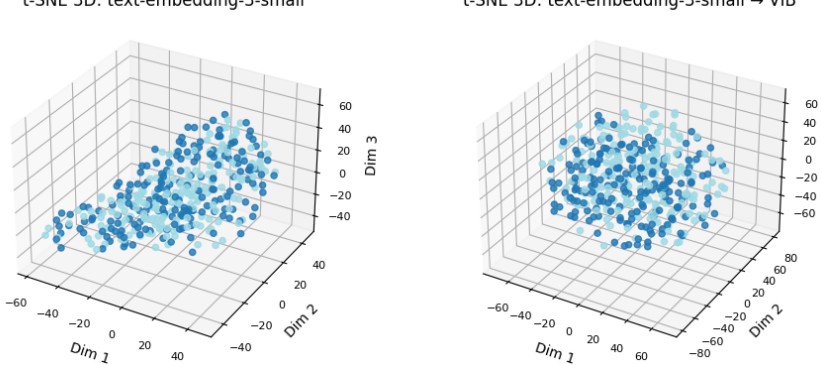

Figure 5: 3D t-SNE projection of 1536-dim segment embeddings for SelfRegulationSCP2 dataset, colored by class. *Left:* raw OpenAI embeddings; *Right:* embeddings after VIB filtering.

Table 4: Per–class performance on the sentiment classification task.

| Class | Precision | Recall | $F_1$–score |
|---|---|---|---|
| Negativity | 0.76 | 0.76 | 0.76 |
| Positivity | 0.71 | 0.71 | 0.71 |
| **Accuracy** | — | — | 0.74 |
| **Macro avg.** | 0.73 | 0.73 | 0.73 |
| **Weighted avg.** | 0.74 | 0.74 | 0.74 |

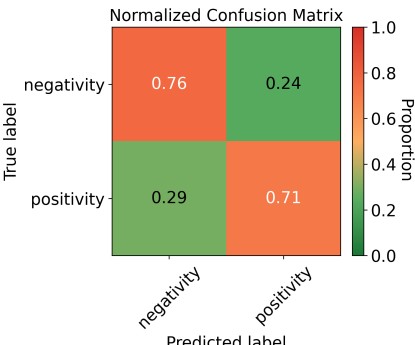

Figure 6: Normalized confusion matrix for the ADEPT v2.0 pipeline on the SelfRegulationSCP2 dataset. Rows correspond to true movement classes (negativity, positivity), columns to predicted classes; cell intensity indicates per–class recall.

## I DETAILED RESULTS ON PREDICTING FUTURE BITCOIN PRICE TREND

Figure 7 shows 2D t-SNE visualizations of 1536-dim segment embeddings from the Bitcoin market dataset, colored by next-day movement class. *Left:* raw text embeddings form elongated, intertwined trajectories with substantial class overlap. *Right:* VIB filtering embeddings produce more homogeneous clusters for positive, negative, and stable days, indicating enhanced discriminability.

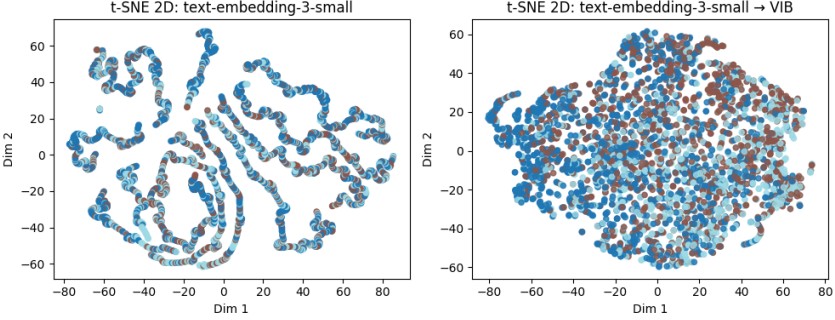

Figure 7: 2D t-SNE projection of text-serialized Bitcoin segment embeddings (1536 D), colored by next-day movement class. *Left:* raw OpenAI embeddings; *Right:* embeddings after VIB filtering.

Table 5 reports per-class precision, recall, and $F_1$-scores for the three movement classes in the Bitcoin price trend test set, alongside overall accuracy (0.88), macro- and weighted-average metrics. The model achieves strong performance across all classes, with highest $F_1$ on the "stable" class.

Table 5: Per-class performance on the Bitcoin market dataset.

| Class | Precision | Recall | $F_1$-score |
|---|---|---|---|
| Long | 0.87 | 0.88 | 0.87 |
| Short | 0.82 | 0.80 | 0.81 |
| Stable | 0.92 | 0.92 | 0.92 |
| **Accuracy** | — | — | 0.88 |
| **Macro avg.** | 0.87 | 0.87 | 0.87 |
| **Weighted avg.** | 0.88 | 0.88 | 0.88 |

Figure 8 shows the normalized confusion matrix for our IB-filtered embedding pipeline on the Bitcoin dataset. Rows correspond to true next-day movement classes (long, short, stable) and columns to predicted classes; cell intensity indicates per-class recall.

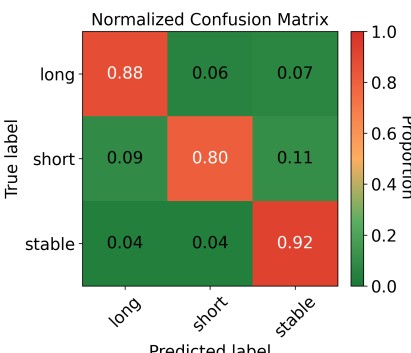

Figure 8: Normalized confusion matrix for the IB-filtered pipeline on the Bitcoin market dataset. Rows correspond to true movement classes (long, short, stable), columns to predicted classes; cell intensity indicates per-class recall.

## J   DETAILED RESULTS ON PREDICTING THE CAUSE OF HYDROPOWER RELIABILITY ISSUES

Figure 9 shows 3D t-SNE projections of these 768-dim segment embeddings colored by cause code: *Left:* raw `nomic-embed-text-v1` embeddings display diffuse, overlapping clusters; *Right:* VIB-filtered embeddings form compact, well-separated clusters, indicating improved discriminability and noise suppression.

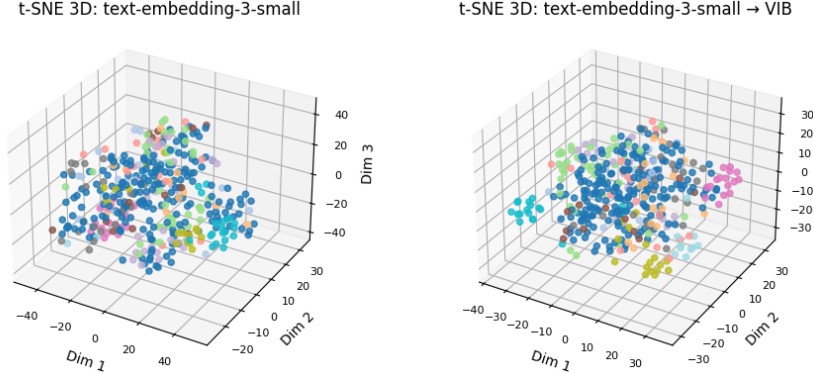

Figure 9: 3D t-SNE projection of 15 min segment embeddings from the HRI dataset, colored by failure cause code (13 classes). *Left:* raw 768-dim Nomic text embeddings; *Right:* embeddings after IB filtering.

Table 6 reports per–cause-code precision, recall, and $F_1$–scores for the 13 failure types in the HRI test set, alongside overall accuracy (0.74), macro– and weighted–average metrics. Perfect $F_1$–scores (1.00) are achieved on several well-represented classes (e.g., 7030, 7050), while rare classes (e.g., 7009) suffer from zero recall.

Figure 10 shows the normalized confusion matrix for our VIB pipeline. Each row is a true cause code and each column the predicted code; cell intensities indicate per-class recall. Notable misclassifications occur between codes 3620 and 3710, reflecting similar pre-failure sensor signatures.

Table 7 reports, for each maintenance event, the model's Top-1 through Top-3 predicted failure-mode classes and their associated probabilities, alongside the actual observed class. While the Top-1 selection yields only 74.4% accuracy, expanding the recommendation to the Top-2 candidates attains 97.5% coverage of the true class. In a maintenance-industry context—where overlooking the true failure mode can have costly consequences—providing a short ranked list of likely failure modes is therefore far more reliable and actionable than a single "best" guess.

Table 6: Per-class performance on the HRI dataset.

| Class | Precision | Recall | F$_1$–score |
|---|---|---|---|
| 3620 | 1.00 | 0.40 | 0.57 |
| 3710 | 0.50 | 0.50 | 0.50 |
| 4560 | 1.00 | 0.50 | 0.67 |
| 7009 | 0.00 | 0.00 | 0.00 |
| 7030 | 1.00 | 1.00 | 1.00 |
| 7050 | 1.00 | 1.00 | 1.00 |
| 7099 | 0.43 | 1.00 | 0.60 |
| 7110 | 0.89 | 0.76 | 0.82 |
| 9696 | 0.75 | 1.00 | 0.86 |
| **Accuracy** | — | — | 0.74 |
| Macro avg. | 0.73 | 0.68 | 0.67 |
| Weighted avg. | 0.85 | 0.74 | 0.76 |

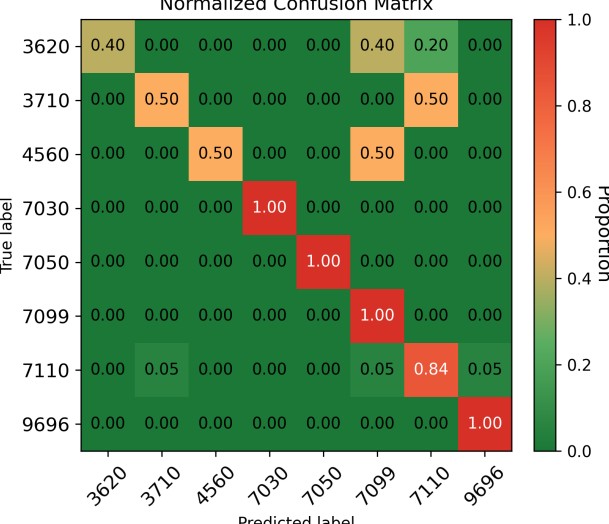

Figure 10: Normalized confusion matrix for the IB-filtered pipeline on HRI. Rows correspond to true failure codes, columns to predicted codes, and cell intensity indicates per-class recall.

Table 7: Predicted class probabilities per event (Top-1 and Top-2 only).

| EventID | Actual | Top-1 Class | Top-2 Class | Top-1 Prob | Top-2 Prob | EventID | Actual | Top-1 Class | Top-2 Class | Top-1 Prob | Top-2 Prob |
|---|---|---|---|---|---|---|---|---|---|---|---|
| 6 | 3620 | 7110 | **3620** | 0.991 | 0.008 | 222 | 7110 | **7110** | 7099 | 0.999 | 0.001 |
| 9 | 3620 | **3620** | 7110 | 0.999 | 0.001 | 230 | 3710 | **3710** | 7099 | 0.999 | 0.000 |
| 17 | 3620 | **3620** | 4600 | 0.851 | 0.143 | 233 | 7110 | **7110** | 7050 | 0.998 | 0.001 |
| 39 | 7030 | **7030** | 4560 | 0.542 | 0.448 | 263 | 7110 | **7110** | 9300 | 0.999 | 0.001 |
| 48 | 4560 | **4560** | 7030 | 0.690 | 0.269 | 306 | 7110 | 9696 | **7110** | 0.523 | 0.404 |
| 78 | 4560 | 7099 | **4560** | 0.999 | 0.001 | 347 | 3620 | 7099 | **3620** | 0.522 | 0.472 |
| 85 | 7050 | **7050** | 4600 | 0.999 | 0.001 | 357 | 7110 | **7110** | 7050 | 0.996 | 0.002 |
| 89 | 7050 | **7050** | 7110 | 0.833 | 0.164 | 367 | 7110 | **7110** | 3620 | 0.999 | 0.001 |
| 139 | 9696 | **9696** | 7050 | 0.999 | 0.001 | 402 | 7110 | **7110** | 9696 | 0.999 | 0.000 |
| 140 | 9696 | **9696** | 7050 | 0.999 | 0.001 | 406 | 7110 | **7110** | 4560 | 0.999 | 0.000 |
| 143 | 9696 | **9696** | 7110 | 0.990 | 0.006 | 410 | 7099 | **7099** | 7009 | 0.999 | 0.0002 |
| 145 | 7110 | 3710 | **7110** | 0.673 | 0.280 | 421 | 7110 | **7110** | 4560 | 0.999 | 0.000 |
| 152 | 7110 | **7110** | 7050 | 0.995 | 0.002 | 445 | 7110 | 7009 | **7110** | 0.821 | 0.144 |
| 163 | 7110 | **7110** | 3620 | 0.997 | 0.001 | 478 | 7110 | **7110** | 9696 | 0.999 | 0.000 |
| 167 | 7110 | **7110** | 4560 | 0.999 | 0.001 | 502 | 3620 | 7099 | **3620** | 0.606 | 0.371 |
| 192 | 7110 | **7110** | 4560 | 0.999 | 0.001 | 554 | 7099 | **7099** | 3710 | 0.999 | 0.000 |
| 202 | 7110 | **7110** | 3710 | 0.959 | 0.037 | 560 | 7110 | 7009 | **7110** | 0.724 | 0.266 |
| 207 | 7099 | **7099** | 7110 | 0.999 | 0.001 | 570 | 3710 | 7110 | **3710** | 0.999 | 0.000 |
| 209 | 7110 | **7099** | 7110 | 0.935 | 0.057 | 571 | 7110 | **7110** | 3710 | 0.879 | 0.116 |
| 215 | 7110 | **7110** | 7009 | 0.999 | 0.001 | | | | | | |

