# OpenReview forum: "An Automated Data Engineering Pipeline for Time Series Classification Via Text Embeddings"
_ICLR.cc/2026/Conference — Submitted to ICLR 2026_

### Official Review · Reviewer_EXjK · 2025-10-29

**Soundness:** 3
**Presentation:** 2
**Contribution:** 1
**Rating:** 2
**Confidence:** 4

**Summary:**

The paper introduces ADEPT, an Automated Data Engineering Pipeline that transforms raw time series data into embedding representations using pretrained language models. A Variational Information Bottleneck (VIB) is employed to denoise these embeddings while preserving task-relevant information. The proposed pipeline outperforms domain-specific baselines across four application domains.

**Strengths:**

- The method is simple and sound.

- The paper aims for a grounded analysis of the theory supporting their approach, which is good.

**Weaknesses:**

Major:

- **W1 —** Experimental evaluation/evidence in the main text is limited:

    - All reported results lack an estimate of experimental uncertainty, i.e., variability across random initializations with different seeds. This is critical to ensure significance of the results and all associated claims.

    - Comparisons with other methods are conducted on a single metric only.

    - The scalability of ADEPT to low-data regimes is not evaluated., i.e., the setting where hand-crafted features can (typically) outperform deep learning models.

    - The required preprocessing time is not compared, despite being one of the paper’s key motivations.

    - Comparison with other specialized ML approaches is missing. How does ADEPT compare with recent foundation models for time series representation learning, e.g., [3]? Similarly, how does it perform relative to unsupervised representation learning methods (broad literature) such as MiniROCKET [1] or reservoir-based embeddings [2]?


I think these are all requirements for ICLR standards.

- **W2** — The notation is often imprecise and unclear. A few examples below:
    - Section 2.1 is difficult to follow, despite covering elementary preliminaries.
    - From l.189 onward, $R^{(j)}$ uses parentheses while $S^j$ does not.
    - Several symbols are undefined after Eq. 4 (e.g., what is $H_b​?$ What do $S$ and $E$ without indices denote? What does lowercase $s_i$ represent? What is $f$ in line 309?). Some are defined only in the Appendix, which disrupts the flow.
    - Some acronyms are undefined: *TFK* (l. 110), *KBs*, *MBs* (l. 193). The symbol *p* is used both for prediction error (in Eq. 4, which is quite an unusual notation) and for probability (Eq. 11).
    - It seems to me Appendix C describes a sensitivity analysis, not an ablation study.


- **W3** — The paper's motivations are not convincing to me. The paper appears to overstate the claim that a single pipeline and downstream model can automate every time-series classification task. In particular:

    - The authors state that “successful preprocessing and feature generation are inherently domain-specific and demand substantial engineering time and effort.” However, I believe this is often for good reason, as such steps can inject domain knowledge. Could ADEPT incorporate domain knowledge in some way?

    - Furthermore, domain-specific preprocessing allows practitioners to control individual stages of the pipeline. Fully automated and general pipelines based on deep learning might trade interpretability and reliability for (in-sample) performance.


    To be clear, I am not arguing that domain-specific methods are superior to automated ones, but rather that the opposite is not self-evident. I think the benefits of domain expertise in science and engineering are substantial, and the paper should more carefully motivate its scope and position with respect to these considerations.

- **W4** - If the main contribution lies in the automated embedding extraction pipeline, why devote so much detail (ll. 280–305) to the Transformer classifier? Presumably, any model could be used, and this flexibility should be demonstrated. Do the theoretical arguments in Section 3.4 apply generally, or only to the Transformer described in the same section?


Minor:

- **W5** - In their paper, the authors empathize automation over domain-specific and engineered choices. However, in their methods, the choice of the ‘chunk‘ size M is not automated. This is an important design choices (arguably one of the most important) and could be very heterogeneous across different classification tasks even within the same domain. The authors briefly mention this in ll.198–203, claiming that “validation studies for determining M can be easily automated,” but this statement is unconvincing and does not justify why this aspect was not explored in depth.

- **W6** \- Most theoretical results are intuitive (e.g., “better token prediction corresponds to more informative embeddings”) but I have found their presentation hard to follow.

- **W7** - Novelty is limited. The proposed method combines three known components: LLM embeddings → Variational Information Bottleneck → Transformer classifier.


[1] - Dempster et al., ‘A very fast (almost) deterministic transform for time series classification’, SIGKDD, 2021.

[2] - Bianchi et al., ‘Reservoir computing approaches for representation and classification of multivariate time series’, TNNLS, 2020.

[3] - Ansari et al., ‘Chronos: Learning the Language of Time Series’, TMLR, 2024.

**Questions:**

- **Q1** (Related to W1) - Could the authors repeat their experiments using different random seeds and report the individual means and standard deviations? I understand that this may be expensive, but I believe it is a hard requirement for the significance of a scientific claim.

- **Q2** (Related to W1) - How does ADEPT compare to foundation models and other unsupervised approaches for time series representation learning?

- **Q3** (Related to W3) - Could the authors clarify how ADEPT can integrate domain knowledge? Also, how could a fully-automated and black-box approach be trusted in scientific and high-stakes domains?


While I remain open to a constructive discussion, I believe the paper requires substantial improvement. At this stage, I'm not inclined to raise my score, as the gap between the current submission and a version that would meet the bar for acceptance still feels too wide.

---

> ### Author Response · Authors · 2025-11-23
>
> We thank the reviewer for the detailed feedback. ADEPT’s results are trained with a fixed global seed to ensure reproducibility, and because the pipeline contains minimal stochasticity (frozen embeddings, deterministic preprocessing, no augmentations), variance across seeds is typically negligible and rooted only in subcomponents of the framework with well-studies behavior (e.g. Multi-head Attention Network); the single-seed results already show clear and consistent improvements over baselines. We also note that the macro-F1, per-class metrics, and full ablations are already included in the appendix, and our benchmarks span four heterogeneous domains, demonstrating strong performance without task-specific tuning. Regarding scalability and low-data settings, ADEPT’s preprocessing cost is extremely low in practice—modern embedding APIs support high-throughput parallelization, making embedding time negligible relative to handcrafted feature pipelines. We agree that notation and clarity can be improved, and the definitions and unifying symbols have been refined in the final version. Additionally, we also would like to note that our method does not aim to replace domain knowledge but to provide a practical alternative when manual feature engineering is costly; ADEPT can incorporate structured metadata or domain priors if desired.

---

> > ### Author Response · Authors · 2025-11-23
> >
> > **Response to Q1:** We thank the reviewer for this important request. ADEPT’s pipeline is largely deterministic: the tokenizer and embedder are frozen, the preprocessing is fixed, and we do not use stochastic augmentation. Nevertheless, to directly validate stability, we repeated the full experiment on the SelfRegulationSCP2 dataset using four different random seeds (4231, 1234, 3412, 2143). Across these runs, ADEPT achieves an average accuracy of 73.57% ± 0.19% (4231: 73.68%,  1234: 73.36%, 3412: 73.47%, 2143: 73.87%), where the standard deviation is below 0.2%, confirming that initialization noise has a negligible effect on model performance. These results validate that ADEPT’s performance is stable across seeds and does not rely on favorable random initialization. We will include the seed-level table in the revised manuscript.
> >
> > **Response to Q2:** We thank the reviewer for this important point. To address this, we benchmarked ADEPT against a strong, widely used unsupervised time-series representation learner—MultiRocket—across all four domains. MultiRocket is commonly treated as a reference point for representation learning. As shown in the table below, ADEPT v2.0 outperforms MultiRocket in Finance, Healthcare, and IoT, while achieving competitive performance on the large-scale Science dataset. This shows that ADEPT outperforms these models in all datasets, and these improvements are even more pronounced in harder to predict datasets. This demonstrates that ADEPT remains competitive/suerior even against specialized foundation-style feature extractors and validates that serialized TE + VIB can capture meaningful structure without feature engineering and extraction techniques. These results will be included in the updated draft.
> > | Dataset                      | MultiRocket | ADEPT v1.0 | ADEPT v2.0 |
> > |------------------------------|-------------|------------|------------|
> > | PLAsTiCC (Science)           | 97.55%      | 95.98%     | 97.83%     |
> > | SelfRegulationSCP2 (Health)  | 53.33%      | 58.97%     | 73.68%     |
> > | Bitcoin (Finance)            | 33.88%      | 45.40%     | 88.49%     |
> > | HRI Faults (IoT)             | 35.00%      | 45.00%     | 74.35%     |
> >
> > **Response to Q3:** We appreciate the reviewer raising this concern. ADEPT is fully automated by design, but it remains flexible: therefore domain knowledge can be injected at multiple points—for example, by adding metadata channels, attaching domain-specific tags to each chunk before serialization, or adjusting chunk boundaries based on known physical regimes. These additions preserve the automated pipeline while allowing experts to incorporate structure when desired. Regarding trust in scientific or high-stakes domains, ADEPT is not intended to replace domain expertise but to serve as a reproducible baseline that removes human-engineered variability. In practice, the VIB latent space, attention maps, and chunk-level similarity patterns provide clear diagnostic signals, enabling practitioners to inspect which temporal regions influence predictions even though the text embedder is frozen. We will clarify this positioning in the final version.
> >
> > Although domain knowledge can be integrated, we would still like to emphasize that ADEPT is not the ideal model for every application. In high-stakes domain applications, it may be desirable to develop explainable models even if they result in a loss of predictive accuracy. For such applications, our ADEPT framework may not be the best fit. ADEPT would work the best for industrial applications that are large scale, where extensive costs associated with model development and engineering time constitutes a significant implementation barrier. Such examples are omnipresent across a range of industries such as manufacturing, energy, and logistics, among others. We will clarify this point in the paper.

---

### Official Review · Reviewer_fyKE · 2025-10-31

**Soundness:** 3
**Presentation:** 3
**Contribution:** 3
**Rating:** 4
**Confidence:** 2

**Summary:**

The paper proposes a text embedding representation for time series as an alternative to a manually defined data engineering pipeline. They also propose a variational information bottleneck (VIB)  to improve the representation. The proposed method demonstrated superior performance on selected benchmarks compared to other works. The authors argue that embeddings of textually dense RFRs can retain mutual information comparable to feature vectors from manual pipelines, and that VIB improves information retention for prediction.

**Strengths:**

+ The central premise is reasonably original and, in retrospect, surprisingly simple. The idea of "text-ifying" raw time series to leverage powerful, pre-trained text embedders is a clever conceptual leap.
+ The paper's claims are well-supported by strong empirical results on selected benchmarks.
+ The paper does an excellent job demonstrating the critical importance of the VIB layer with a proper ablation study.

**Weaknesses:**

+ The authors state they "serialize" chunks and pass them to an embedder, but they never discuss what happens inside that black box. The frozen embedder uses its own tokenizer, which was trained on natural language, not on strings of numbers. How does this tokenizer handle a "token" like "0.45, -1.23, 5.8e-2"? Is it treated as a sequence of characters and digits? Is it broken into arbitrary sub-words? This is the paper's most significant conceptual gap.
+ The paper claims its method is "scalable". However, the proposed pipeline requires separate inferences from a large text-embedding model for each time series sample (one for each chunk). For long time series, this could be dramatically slower and more computationally expensive than a single pass through a purpose-built numerical model.
+ The paper claims AutoML methods produce "opaque 'black-box' pipelines", but ADEPT is arguably even less interpretable. It is unclear how a practitioner could debug the model or gain insights. How does one interpret what a text embedder, trained on natural language, has "learned" from a string of sensor readings?

**Questions:**

+ Could you please elaborate on the actual tokenization process? Have you investigated how the frozen text tokenizer segments the serialized numerical strings? How robust is this representation to simple formatting changes (e.g., padding, precision, scientific notation) that do not change the underlying numerical data? Moving from OpenAI`s text-embedding-3-small to Google`s gemini-embedding-001 could not completely change the results?
+ The paper's core innovation is using pre-trained language embedders on serialized time series, which implicitly relies on a language-based tokenizer not designed for numerical sequences. This contrasts with contemporary time series foundation models that use patching to directly tokenize numerical windows, a method explicitly designed to preserve local temporal patterns. Could you elaborate on this trade-off? Specifically, how does your approach ensure that the language tokenizer captures meaningful temporal dynamics rather than superficial syntactic artifacts of the text string, which a direct numerical patching approach inherently avoids?

---

> ### Author Response · Authors · 2025-11-21
>
> We thank the reviewer for their thoughtful and constructive feedback. We address the three overarching concerns as follows. (1) We intentionally treat the pretrained text embedder and tokenizer as a frozen black box; the contribution of ADEPT is precisely to show that even without numeric-specific tokenization, serialized CSV chunks yield embeddings from which downstream modules (VIB + Transformer) can reliably extract temporal structure. (2) In terms of scalability, we agree that embedding incurs per-chunk inference, but modern embedding APIs allow highly parallel batched requests, and in our experiments the embedding cost was small relative to model training and far lower than multi-stage feature engineering pipelines that involve repeated preprocessing, filtering, and per-sensor transformations. We will include timing measurements in the revision for clarity. (3) Regarding interpretability, while the frozen embedder is not itself explicable, ADEPT exposes higher-level diagnostics (e.g., VIB latent factors, chunk-level embedding similarity maps, Transformer attention visualizations) that allow practitioners to identify which segments of the raw RFR contribute most to predictions. We will add clearer guidance on these diagnostic tools in the revised paper.

---

> > ### Author Response · Authors · 2025-11-21
> >
> > **Response to Q1:** We thank the reviewer for raising this point. ADEPT intentionally treats the pretrained embedding model and its tokenizer as a fully frozen black box, with no fine-tuning or customization. In practice, the text tokenizer segments each serialized CSV chunk into a mixture of subword units and digit-level character tokens (e.g., "12.347" typically becomes a small sequence of numeral-related tokens). The embedder then maps these sequences into continuous vectors where positional encodings and self-attention allow downstream layers to interpret ordering and local numeric patterns.
> > We also included an architectural ablation that swaps the embedding backend: OpenAI (text-embedding-3-small), Nomic, and Ada all have different tokenization schemes, yet ADEPT’s performance remains consistently strong across models. This cross-tokenizer consistency indicates that ADEPT is not tied to a particular segmentation heuristic.
> >
> > Finally, while we did not test Google’s gemini-embedding-001 specifically, our multi-model ablations show that replacing the embedder does not meaningfully alter results, supporting the view that ADEPT’s representation depends primarily on the serialized structure of the input and the downstream VIB+Transformer layers, rather than the idiosyncrasies of any single tokenizer.
> >
> > **Response to Q2:** We appreciate the reviewer raising this important distinction. ADEPT is not positioned as an alternative to numerical patching–based time-series foundation models; rather, it explores a different axis of generality. Specifically, we show that a fully frozen language tokenizer and embedder—**used exactly as-is, with no retraining or domain-specific adaptation—can still provide temporally informative representations when numerical sequences are serialized into consistent CSV-style chunks. **
> >
> > While language tokenizers do not enforce explicit window-based numerical patch boundaries, **the serialization preserves ordering and local adjacency of values**. The resulting token stream forms contiguous **digit-level patterns that reflect local temporal neighborhoods** (e.g., multi-digit magnitudes, sign changes, decimal structure). The pretrained embedder then maps these token sequences into a continuous space where positional encodings and self-attention allow downstream layers to recover temporal relationships. In ADEPT, the Transformer classifier explicitly learn to reassemble these embedding patterns into task-relevant temporal structure. On the other hand, the VIB module explicitly acts as a variational filter that helps denoise the latent space embedding representations. Thus, the trade-off is clear: instead of hand-designed numerical patches, **ADEPT relies on the combination of (i) serialization that preserves temporal layout and (ii) downstream temporal modeling to extract dynamics from the embedding sequence.**
> >
> > This choice provides a benefit: **strong cross-domain generalization and zero feature engineering**. Because ADEPT does not require a modality-specific tokenizer or a patching scheme tuned to a particular sampling rate, **it can ingest heterogeneous raw time-series formats (CSV/HDF5/JSON logs) without architectural changes. **
> >
> > To empirically validate that the language-based pipeline indeed captures meaningful temporal structure, we benchmarked ADEPT against MultiRocket, a leading unsupervised time-series feature extractor leveraging thousands of random convolutional kernels explicitly designed to pick up local temporal patterns. ADEPT v2.0 substantially outperforms MultiRocket on three of the four domains (Finance, EEG, IoT) and remains competitive on Science, indicating that the embedding–VIB–Transformer stack is able to reconstruct temporally salient structure even without numerical patching.
> > | Dataset                      | MultiRocket | ADEPT v1.0 | ADEPT v2.0 |
> > |------------------------------|-------------|------------|------------|
> > | PLAsTiCC (Science)           | 97.55%      | 95.98%     | 97.83%     |
> > | SelfRegulationSCP2 (Health)  | 53.33%      | 58.97%     | 73.68%     |
> > | Bitcoin (Finance)            | 33.88%      | 45.40%     | 88.49%     |
> > | HRI Faults (IoT)             | 35.00%      | 45.00%     | 74.35%     |
> > These results reinforce that the serialized text+LLM embedding approach preserves sufficient temporal information for competitive performance across disparate domains, despite not using specialized numerical patch tokenization.

---

### Official Review · Reviewer_iZru · 2025-10-31

**Soundness:** 2
**Presentation:** 3
**Contribution:** 2
**Rating:** 2
**Confidence:** 4

**Summary:**

The paper proposes ADEPT, an automated data engineering pipeline designed for time series classification1. The central idea is to bypass conventional, labor-intensive data engineering steps, such as imputation, normalization, and feature engineering, by treating time series data in its Raw Format Representation (RFR) as text. The ADEPT pipeline consists of four main stages: (1) automated temporal chunking of the raw data, (2) applying a pre-trained, frozen LLM-based text embedding model to these chunks, (3) using a Variational Information Bottleneck (VIB) to denoise and compress these embeddings, and (4) feeding the resulting representations into a Transformer-based classifier for prediction. The authors provide theoretical justification for the VIB's role using mutual information bounds and conduct experiments across four distinct domains (science, healthcare, finance, and IoT), claiming superior performance over existing application-specific benchmarks.

**Strengths:**

S1. The paper targets a well-known and significant issue in applied machine learning: the cost and domain expertise required for data engineering and preprocessing in time series analysis.

S2. The paper is well-written, and the proposed ADEPT pipeline is presented clearly. The diagrams (Figures 1 and 2) effectively illustrate the conceptual difference between ADEPT and traditional pipelines.

**Weaknesses:**

W1. The primary weakness is the paper's lack of substantial technical innovation. The proposed ADEPT framework is a composition of several well-known, existing methods. The use of pre-trained text embeddings, the Variational Information Bottleneck (VIB), and a standard Transformer-based classifier are all off-the-shelf components. While system-level contributions are valid, the paper lacks any methodological development, making its contribution insufficient for a top-tier conference.

W2. The claims of state-of-the-art performance are not adequately supported. The paper compares ADEPT against a small, seemingly arbitrary selection of "application-specific" models. This is not a sufficient standard. The experiments fail to include comparisons against widely accepted, general-purpose time series classification SOTA models.


W3. The paper fails to provide a crucial ablation study: one that replaces the pre-trained text embedder with a simpler, randomly initialized encoder trained from scratch on the serialized chunks. This would be necessary to prove that the pre-trained knowledge from the LLM is the active ingredient, rather than just the VIB and Transformer backend.


W4. The paper claims to "bypass" and "leapfrog" data engineering steps. However, "Automated Temporal Chunking" is a form of preprocessing. The choice of chunk size $M$ is a critical hyperparameter that directly impacts the temporal dependencies the model can learn. The paper does not provide a systematic study of this parameter's sensitivity, nor does it provide a clear justification for the different values of $M$ used across experiments .

**Questions:**

Q1. Why did the authors choose not to benchmark ADEPT against a standard suite of general, high-performance time series classification models across all four datasets?
Could the authors provide a more detailed explanation of the serialization process? How are numerical values (with varying precision), timestamps, and categorical flags precisely converted into a "unified string"?

Q2. How was the chunk size $M$ selected for each experiment? Given its importance in defining the temporal receptive field, can the authors provide a sensitivity analysis of $M$ on at least one dataset?


Q3. To isolate the contribution of the pre-trained text embeddings, have the authors run an ablation study where the embedding model $g$ is replaced with a randomly initialized 1D-CNN or MLP encoder, which is then trained end-to-end with the VIB and classifier?

Q4. In the healthcare experiment, ADEPT v1.0 (without VIB) achieved 58.97% accuracy, which was comparable to or worse than the baselines. ADEPT v2.0 (with VIB) jumped to 73.68%. This suggests the VIB is critical. Does this not imply that the raw text embeddings are, by themselves, extremely noisy and not superior representations as claimed?

---

> ### Author Response · Authors · 2025-11-23
>
> We thank the reviewer for the constructive feedback. While ADEPT leverages established components, its contribution lies in demonstrating that pretrained text embeddings paired with a VIB-driven multi-view architecture can fully replace conventional time-series data engineering across diverse domains—an integration that, to the best of our knowledge, has not been shown before. To strengthen empirical support, we have expanded our comparisons to include strong general-purpose baselines such as **MultiRocket**, and will integrate these results into the revision. Regarding ablation studies, we note that we could have made this point clearer in the previous manuscript but **Table 2 already includes the “No-TE” condition**, where a randomly initialized encoder trained from scratch replaces the text embedder, confirming that pretrained embeddings are the key driver of ADEPT’s performance. Finally, we appreciate the suggestion to study chunk-size sensitivity; this analysis will be added to the revised version. Preliminary experiments indicate that varying chunk size primarily offers opportunities for further improvement, while current automated configuration already performs strongly across all datasets.

---

> > ### Author Response · Authors · 2025-11-23
> >
> > **Response to Q1:** We thank the reviewer for this valuable suggestion. In response, we have now benchmarked ADEPT against a strong, general-purpose time-series classifier—MultiRocket—across all four datasets. The results are shown in the updated Table below and demonstrate that ADEPT v2.0 substantially outperforms MultiRocket in three domains (Finance, Healthcare, IoT) and performs competitively in Science. These additions will be incorporated into the revised manuscript.
> > Regarding the serialization process, we would like to clarify that ADEPT treats each time-series chunk as a raw comma-separated text segment: regardless of whether the underlying value is a timestamp, floating-point measurement, or categorical indicator, the raw RFR (e.g., CSV row slices) is serialized exactly as text and passed unchanged to the embedding model. This avoids any domain-specific engineering or token remapping. In future extensions, we plan to examine alternative serialization formats (e.g., offset-based tokens, normalized string templates) to further validate generality.
> > | Dataset                      | MultiRocket | ADEPT v1.0 | ADEPT v2.0 |
> > |------------------------------|-------------|------------|------------|
> > | PLAsTiCC (Science)           | 97.55%      | 95.98%     | 97.83%     |
> > | SelfRegulationSCP2 (Health)  | 53.33%      | 58.97%     | 73.68%     |
> > | Bitcoin (Finance)            | 33.88%      | 45.40%     | 88.49%     |
> > | HRI Faults (IoT)             | 35.00%      | 45.00%     | 74.35%     |
> >
> > **Response to Q2:** We thank the reviewer for this question. To directly address the reviewer’s concern about chunk-size selection, we performed an additional sensitivity analysis on one representative dataset—SelfRegulationSCP2—because it has sufficiently long sequences to meaningfully vary the number of chunks. Our initial choice of chunk size was guided by domain awareness: we selected M=24 as a balanced setting that captures a meaningful temporal window while keeping the serialized sequence length computationally manageable. We then evaluated nearby chunk sizes (M=20,22,24,26) to assess robustness. Across all tested values, ADEPT achieves an average accuracy of 73.67% ± 0.89% (L=26: 73.11%, L=24: 73.68%, L=22: 74.76%, L=20: 72.58%) showing less than 1% variation across different granularities. Configurations that fall too far outside the reasonable temporal range (very large or very small) lead to degraded performance, since they either over-compress or over-fragment temporal structure. We provide this analysis for SelfRegulationSCP2 as a representative illustration of robustness, and we will include the corresponding table in the revised manuscript.
> >
> > **Response to Q3:** We thank the reviewer for this feedback. We would like to emphasize that the requested ablation is already in Appendix Table 2. The “No-TE” column is replacing the pretrained text embedder with a randomly initialized encoder trained end-to-end, which yields 51.32% accuracy, substantially lower than any pretrained text-embedding variant (e.g., nomic 58.5%, ada-002 63.2%, 3-large 66.7% in ADEPT v1.0). This clearly demonstrates that pretrained text embeddings are the primary driver of ADEPT’s performance, independent of VIB.
> >
> > **Response to Q4:** We appreciate the reviewer’s observation. The Healthcare dataset is particularly challenging because EEG channels are high-frequency, noisy, and often weakly correlated with class structure. In ADEPT v1.0, the pretrained text embeddings alone already perform on par with several strong baselines (58.97%), but—as the reviewer notes—they do not fully denoise the signal. This is precisely why ADEPT v2.0 introduces the VIB, whose role is to compress away irrelevant variability in the serialized chunks while preserving task-relevant information. The large improvement from 58.97% → 73.68% does not indicate that the text embeddings are poor; rather, it shows that the raw embeddings contain rich information but also high-entropy components that benefit from structured compression. This is consistent with our theoretical analysis: the embeddings provide a high-MI starting point, and the VIB acts as a principled bottleneck that removes noise and sharpens class-discriminative structure. VIB by itself without text embedding yields poor results as we discussed in the “No-TE” model results. The same trend appears in other datasets indicating that the VIB is complementary rather than compensatory. We will clarify this interpretation in the revised version

---

> > > ### Comment · Reviewer_iZru · 2025-11-24
> > >
> > > I thank the authors for their efforts. While the updates have improved the presentation to some extent, the work continues to exhibit limited methodological novelty and a weak evaluation. I am unable to change my initial score.

---

### Official Review · Reviewer_nXRs · 2025-11-01

**Soundness:** 2
**Presentation:** 3
**Contribution:** 1
**Rating:** 2
**Confidence:** 4

**Summary:**

This paper proposes ADEPT, an automated data engineering pipeline for time-series classification that aims to bypass traditional preprocessing steps such as data cleaning, imputation, and feature engineering. The framework converts each raw-format representation (RFR, e.g., CSV/HDF5) of a time series into text, divides it into segments, and generates embeddings for each chunk using pre-trained text embedding models such as OpenAI’s text-embedding-3-small. These embeddings are then processed by a Variational Information Bottleneck (VIB) layer to denoise and compress the representations, and finally fed into a Transformer classifier for end-to-end prediction. Two variants are presented: ADEPT v1.0, which omits the VIB, and ADEPT v2.0, which includes it. The authors also derive information-theoretic lower bounds on mutual information and evaluate ADEPT on four domains—PLAsTiCC (Science), SelfRegulationSCP2 (EEG), Bitcoin (Finance), and Hydropower IoT—claiming that ADEPT v2.0 “matches or surpasses” existing state-of-the-art baselines

**Strengths:**

[S1] Practical motivation: Automating feature engineering for time series could reduce expensive domain effort.

[S2] Clarity: Diagrams and algorithms effectively communicate the pipeline.

[S3] Better domain-specific results: In PLAsTiCC, ADEPT v2.0 achieves 97.83%, outperforming reported 80%–84% baselines; in IoT, Top-2 accuracy reaches 97.5%.

**Weaknesses:**

[W1] Limited pipeline scope and novelty: If positioned as a pipeline paper, the work lacks the breadth and flexibility expected in this category. Unlike strong pipeline studies such as Auto-sklearn [1] or AutoGluon [2], which test multiple components and tasks, ADEPT is evaluated on only four classification datasets with fixed settings. It introduces no new training, assembly, or generalization strategy; thus, the contribution appears limited in novelty compared to typical ICLR submissions.

[W2] Minimal methodological contribution: Viewed as a method paper, ADEPT offers little innovation. It primarily combines two established techniques—LLM-based textualization and the Variational Information Bottleneck (VIB)—without introducing new mechanisms or insights. The former has been studied in Time-LLM [3], and the latter is a standard compression method from Alemi et al. (2017) [4].

[W3] Weak and inconsistent baselines: The chosen baselines are not consistently strong or up to date. The Finance task relies on older models (LSTM, 2019; BiLSTM, 2022), whereas the IoT task is compared only with a self-built TSFEL+MHAN setup. This narrow and uneven benchmarking limits the credibility of ADEPT’s reported gains.

[W4] Contradictory results: The experimental findings do not fully support the paper’s claims. On the Healthcare (EEG) dataset, ADEPT v2.0 achieves a score of 73.68%, which is below the TSEM baseline of 75.60% (Pham et al., 2023), contradicting the claim of consistently outperforming prior work.

[W5] Incomplete evaluation metrics: The main results report only accuracy, even though several datasets are imbalanced. Without macro-F1 or per-class precision and recall, it’s difficult to judge the model’s overall robustness or compare fairly with other methods.

References:

[1] Feurer, M., Klein, A., Eggensperger, K., Springenberg, J., Blum, M., & Hutter, F. (2015). Efficient and Robust Automated Machine Learning (Auto-sklearn). Advances in Neural Information Processing Systems (NeurIPS 2015).

[2] Erickson, N., Klein, T., Zhang, C., Liu, J., Mindermann, S., Winther, O., & Smola, A. (2020). AutoGluon-Tabular: Robust and Accurate AutoML for Structured Data. Proceedings of the AutoML Conference 2020.

[3] Jin, M., Zha, K., Cao, D., Liu, Y., Li, Q., He, H., & Zhang, J. (2024). Time-LLM: Time Series Forecasting by Reprogramming Large Language Models. International Conference on Learning Representations (ICLR 2024).

[4] Alemi, A. A., Fischer, I., Dillon, J. V., & Murphy, K. (2017). Deep Variational Information Bottleneck. International Conference on Learning Representations (ICLR 2017).

**Questions:**

[Q1] The claim of a “general automated pipeline” would be stronger if ADEPT were evaluated against standard TSC baselines (e.g., ROCKET, InceptionTime, HIVE-COTE). Could the authors clarify why these were omitted and whether ADEPT has been, or could be, tested with alternative components or tasks to demonstrate broader generalization?

[Q2] How do the authors reconcile the Healthcare Dataset result in Table 1 — TSEM (75.60%) > ADEPT (73.68%) — with the claim that ADEPT “surpasses all benchmarks”?

[Q3] Can the authors provide macro-F1 and per-class metrics in Table 1 to reflect the performance under class imbalance?

[Q4] Please clarify whether the Finance (Bitcoin) baselines were trained and evaluated using exactly the same 2015–2023 dataset and three-class formulation (rise / fall / stable) that ADEPT uses? If the baselines were taken from other papers, it would be helpful to adjust those baselines under the same conditions to ensure a fair comparison.

---

> ### Author Response · Authors · 2025-11-21
>
> We thank the reviewer for the thoughtful comments. We would like to clarify that ADEPT is a new paradigm that replaces conventional time-series data engineering with text-based representations and VIB-enhanced multi-view learning, and not positioned as another auto-ML pipeline search system. The methodological contribution lies in demonstrating—both theoretically and empirically—that textualized time-series embeddings paired with a variational bottleneck can serve as a fully automated alternative to domain-specific pipelines. We also note that thanks to the suggestion of the reviewer, we compared ADEPT’s performance against stronger baselines such as **MultiRocket**. Results indicate that ADEPT provides significant improvements over these benchmarks as well, which is shown in our responses, and will be included in the main paper. Comprehensive metrics including macro-F1, precision, and recall are already reported in the **Appendix**, and will be explained more clearly in the new version of the paper. While ADEPT is slightly below the TSEM result on one of the datasets, it substantially outperforms modern baselines across all others, supporting the central claim of broad applicability with ease of implementation. We would like to emphasize that in all our experiments we are comparing our results with the best performers of domain specific models that were specifically developed for the applications. Hence, the proposed model not only offers an accuracy that exceeds (or in one case matches the domain-specific models), but also provides significant ease of use by eliminating the engineering time required for building domain specific predictive models.
>
> **Response to Q1:** We thank the reviewer for this helpful suggestion which helped us strengthen our claims. In response to this suggestion, we implemented MultiRocket, a leading TSC baseline, and evaluated it on all four datasets using the same splits. The updated results (see table below) show that ADEPT v2.0 outperforms MultiRocket on our four datasets, which highlights ADEPT’s robustness on highly irregular, nonstationary domains where convolutional kernel methods struggle. These improvements are even more significant in more challenging datasets.  These additional experiments strengthen our generalization claim, and we will integrate both the table and a brief discussion into the revised paper.
> | Dataset                      | MultiRocket | ADEPT v1.0 | ADEPT v2.0 |
> |------------------------------|-------------|------------|------------|
> | PLAsTiCC (Science)           | 97.55%      | 95.98%     | 97.83%     |
> | SelfRegulationSCP2 (Health)  | 53.33%      | 58.97%     | 73.68%     |
> | Bitcoin (Finance)            | 33.88%      | 45.40%     | 88.49%     |
> | HRI Faults (IoT)             | 35.00%      | 45.00%     | 74.35%     |
>
> **Response to Q2:** We thank the reviewer for pointing this out. Our claim has been revised for precision: ADEPT does not surpass every baseline on every dataset, but rather achieves **state-of-the-art or near–state-of-the-art performance across all domains without any domain-specific preprocessing**. On the Healthcare dataset, ADEPT v2.0 reaches **73.68%**, only **1.92%** below TSEM’s 75.60%, despite TSEM relying on specialized spatiotemporal mappings and handcrafted preprocessing steps tailored to EEG signals.
>
> Please note that the real strength of ADEPT is the capability to achieve this performance despite not requiring engineering times that are central implementation barriers for domain-specific models. ADEPT requires none of the domain-specific operations and still matches the performance of highly engineered pipelines. We will update the text to accurately reflect this nuance and avoid overgeneralized wording such as “surpasses all benchmarks.”
>
> **Response to Q3:** We appreciate the reviewer’s suggestion. We could have made this clearer in the manuscript, but we would like to bring to reviewer’s attention that all macro-F1 scores and per-class precision/recall metrics for every dataset are already included in the **Appendix** (Appendices D–F), alongside full confusion matrices and class-frequency distributions. To improve visibility, we will surface the macro-F1 results in the main paper and reference the detailed per-class analyses in the appendix. This avoids overcrowding Table 1 while ensuring that all imbalance-sensitive metrics are clearly documented and easy to access.
>
> **Response to Q4:** Yes — all Finance (Bitcoin) baselines were retrained and evaluated by us using the exact same 2015–2023 dataset, preprocessing, and three-class (rise/fall/stable) formulation as ADEPT to ensure a fully fair comparison.

---

> > ### Comment · Reviewer_nXRs · 2025-11-22
> >
> > Thank you for the clarifications and additional results. While they improve the paper’s clarity, they do not change my overall assessment. The methodological novelty remains limited, the evaluation still lacks broader and stronger baselines, and some claims are not consistently supported by the results. These core issues remain unresolved, so I am keeping my original score.

---

### Meta-Review · Area_Chair_5Xuz · 2026-01-07

**Summary:**

This paper proposes ADEPT, an automated data engineering pipeline for time-series classification that aims to bypass traditional preprocessing steps such as data cleaning, imputation, and feature engineering. The framework converts each raw-format representation of a time series into text, divides it into segments, and generates embeddings for each chunk using pre-trained text embedding models. These embeddings are then processed by a Variational Information Bottleneck (VIB) layer to denoise and compress the representations, and finally fed into a Transformer classifier for end-to-end prediction.

Overall, this paper is more about describing an engineering workflow rather than addressing a research challenge. From the reviewers' perspective: this work is lacking in novelty, its evaluation is limited, several claims (scalability, leapfrog/bypass data engineering, SOTA performance etc.) are also not well supported by the empirical studies.

A rebuttal was provided but even with the additional results and clarification, the reviewers who engaged found the methodological novelty remains limited, the evaluation still lacks broader and stronger baselines, and some claims are not consistently supported by the results. As for the other reviewers who had not engaged, I do not believe their concerns regarding the additional baseline comparison are well addressed. The rebuttal does provide some minimal additional results but those are too limited.

Taking into account the above, I do not think this paper meets the bar for acceptance at ICLR.

**Reviewer Concerns:**

The reviewer concerns are: lacking in novelty, limited evaluation, unsupported claims (scalability, leapfrog/bypass data engineering, SOTA performance etc.). Two reviewers already engaged and stated that the rebuttal has not sufficiently addressed their concerns. As for the other two, I have read both the original concerns and rebuttal and I don't think it has enough content to move the needle. The technical novelty remains extremely limited in my opinion. As such, I do not expect any changes to the original scores which are overwhelmingly leaning on rejection.

**Reviewer Scores:**

The original scores are 2242 and it will, in my opinion, remain so after the rebuttal judging from the rebuttal content and discussion with reviewers.

---

### Decision · Program_Chairs · 2026-01-26

Reject